# VAnim: Rendering-Aware Sparse State Modeling for Structure-Preserving Vector Animation

**Guotao Liang** [1]   **Zhangcheng Wang** [2]   **Chuang Wang** [1]   **Juncheng Hu** [1]   **Haitao Zhou** [1]   **Junhua Liu** [3]   **Jing Zhang** [1]   **Dong Xu** [4]   **Qian Yu** [1 †]

**Prompt:** "*The strings were plucked, and the notes appeared and disappeared.*"

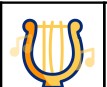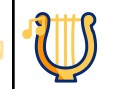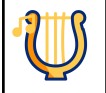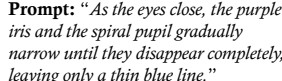

**Prompt:** "*The pen draws a wavy line, and the portrait progress bar fills up.*"

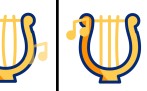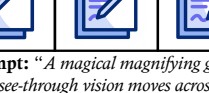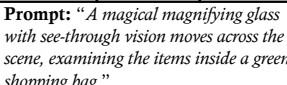

**Prompt:** "*From left to right, the orange balls inside the game console fall one by one.*"

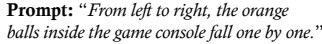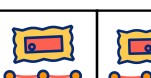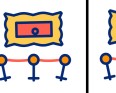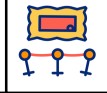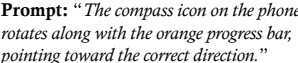

**Prompt:** "*The orange progress bar finishes loading just as the three playing cards come together in a stack.*"

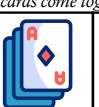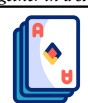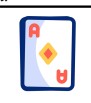

**Prompt:** "*As the eyes close, the purple iris and the spiral pupil gradually narrow until they disappear completely, leaving only a thin blue line.*"

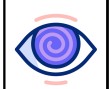

**Prompt:** "*A magical magnifying glass with see-through vision moves across the scene, examining the items inside a green shopping bag.*"

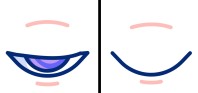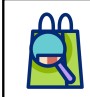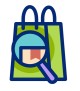

**Prompt:** "*The compass icon on the phone rotates along with the orange progress bar, pointing toward the correct direction.*"

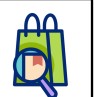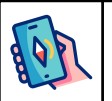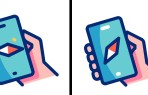

**Prompt:** "*The toothpaste remains stationary; the toothbrush is inserted into the cup and positioned correctly.*"

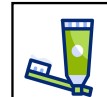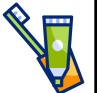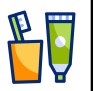

*Figure 1.* **VAnim Generation Gallery.** We present diverse vector animations generated from prompts. The results demonstrate VAnim's capability to execute complex semantic instructions, ranging from **non-rigid deformation** and **sequential logic** to **multi-object interaction**, all while maintaining strict topological consistency. Please refer to the supplementary webpage for full video demonstrations.

## Abstract

Scalable Vector Graphics (SVG) animation generation is pivotal for professional design due to their structural editability and resolution independence. However, this task remains challenging as it requires bridging discrete code representations with continuous visual dynamics. Existing optimization-based methods often destroy topological consistency, while general-purpose LLMs rely on rigid CSS/SMIL transformations, failing to model geometry-level non-rigid deformations. To address these limitations, we present **VAnim**, the first LLM-based framework for open-domain text-to-SVG animation. We reconceptualize animation not as sequence generation, but as *Sparse State Updates* (SSU) on a persistent SVG DOM tree. This paradigm compresses sequence length by over $9.8\times$ while preserving the SVG DOM structure and non-participating elements by construction. To enable precise control, we propose an *Identification-First Motion Planning* mechanism that grounds textual instructions in explicit visual entities. Furthermore, to overcome the non-differentiable nature of SVG rendering, we employ *Rendering-Aware Reinforcement Learning* via Group Relative Policy Optimization (GRPO). By leveraging a hybrid reward from a state-of-the-art video perception encoder, we align discrete code updates with high-fidelity visual feedback. We also introduce **SVGAnim-134k**, the first benchmark for vector animation. Extensive experiments demonstrate that VAnim significantly outperforms state-of-the-art baselines in semantic alignment and structural validity, with additional appendix metrics further validating motion quality and identity preservation.

## 1. Introduction

In the professional landscapes of user interface (UI) design, web engineering, and digital iconography, *Scalable Vector Graphics* (SVG) (Consortium, 1999; Quint, 2003) stand as the de facto standard. Celebrated for their resolution independence, compact file sizes, and, most critically, *structural editability*, SVGs allow designers to precisely manipulate geometric components across devices. Beyond static im-

[1]School of Software, Beihang University, Beijing, China [2]4Paradigm [3]College of Computer Science and Technology, Zhejiang University, Hangzhou, China [4]Department of Computer Science, The University of Hong Kong, Hong Kong, China. Correspondence to: Qian Yu <qianyu@buaa.edu.cn>.

*Proceedings of the $43^{rd}$ International Conference on Machine Learning*, Seoul, South Korea. PMLR 306, 2026. Copyright 2026 by the author(s).

agery, **vector animation** (Dalstein et al., 2015; Mo et al., 2024) plays a pivotal role in modern digital experiences, breathing life into static icons through micro-interactions, loading indicators, and narrative illustrations (Saffer, 2013).

However, automating the creation of such animations remains a formidable challenge, lagging significantly behind other generative domains. While the AI revolution has demonstrated remarkable capabilities in pixel-based video synthesis (Gao et al., 2025; Kong et al.; Wan et al., 2025a) and sparked a surge of interest in static vector generation (Liang et al., 2026; Rodriguez et al., 2025a;b; Xing et al., 2025; 2023; 2024; Yang et al., 2025; Wu et al., 2023; Frans et al., 2022; Vinker et al., 2022; Song et al., 2023; Jain et al., 2023; Hu et al., 2025; Iluz et al., 2023; Zhang et al., 2024a; Thamizharasan et al., 2024; Hu et al., 2024; Wang et al., 2025a; Tang et al., 2024; Wang et al., 2025b; Hu et al., 2026), the intersection of these fields, **text-to-SVG animation** (Gal et al., 2024; Liang et al., 2025; Liu et al., 2025), remains largely unexplored. Pixel-based videos, despite realism, lack the editability required for UI integration. Conversely, existing static vector models focus on spatial composition, failing to model the temporal coherence required for motion. Consequently, creating vector animations currently demands labor-intensive manual keyframing (Tseng et al., 2024).

The difficulty stems from a severe *representation mismatch* when extending static paradigms into the temporal domain. While static SVG generation is often framed as a code synthesis task, a naive application of autoregressive Large Language Models (LLMs) (Bai et al., 2025; Vaswani et al., 2017) to generate frame-by-frame SVG code encounters two prohibitive obstacles: **context explosion** and **structural drift**. Since SVG syntax is verbose, repeating the full document for each animation frame rapidly exceeds the context window (Dai et al., 2019; Beltagy et al., 2020). More critically, regenerating static attributes at every timestep introduces stochastic inconsistencies, causing object identities to flicker or collapse over time, a phenomenon we refer to as *identity drift* (Yuan et al., 2025b; Zhou, 2024).

Existing alternatives fail to adequately address these challenges due to inherent methodological limitations. First, optimization-based approaches like LiveSketch (Gal et al., 2024) rely on iterative differentiable rasterization. However, this paradigm suffers from two critical flaws: *(i) Prohibitive Latency*: the reliance on hundreds of SDS steps results in minute-level generation, precluding interactive applications; *(ii) Topological Instability*: by treating vectors as independent strokes without structural awareness, these methods fail to maintain closed shapes or occlusions, limiting their utility to sparse sketches rather than professional designs.

Second, contemporary Large Language Models (e.g., GPT-5.2 (OpenAI, 2025), Gemini 3 Pro (Google Cloud,

2025)) and declarative animation frameworks such as Keyframer (Tseng et al., 2024) exhibit a strong inductive bias toward *rigid motion*. To avoid explicit coordinate-level reasoning, these systems predominantly synthesize animations using CSS or SMIL transforms (W3C, 2019; SVG Working Group, 2025). However, such representations are mathematically restricted to affine transformations, including translation, rotation, and uniform scaling. This constraint introduces a severe expressive bottleneck: non-rigid deformations, such as waving flags, morphing droplets, or organic character motion, require direct manipulation of path geometry (i.e., the d attribute in SVG path definitions), which lies beyond the representational capacity of affine transforms. Consequently, true shape plasticity and geometry-level temporal evolution remain out of reach for existing approaches.

To bridge this gap, we propose **VAnim**, the first LLM-based framework for open-domain text-to-SVG animation. Our key insight is to model animation as *Sparse State Updates* (SSU) on a persistent *Document Object Model* (DOM) (W3C, 2020), rather than independently generating full frames. VAnim predicts sparse, identifier-anchored updates to a few attributes (e.g., d, transform) instead of rewriting the entire SVG at each timestep. This reduces the effective sequence length by more than $9\times$, mitigating context explosion while preserving the persistent DOM structure and non-participating elements by construction.

Generating such updates requires structured reasoning over both semantics and geometry. We therefore introduce *Identification-First Motion Planning*: inspired by Chain-of-Thought (CoT) reasoning (Wei et al., 2022), the model first links semantic entities to SVG IDs and plans their temporal behavior before emitting code-level edits. This grounds motion in persistent structural components rather than risking structural dissociation or identity loss.

While Supervised Fine-Tuning (SFT) ensures syntactic correctness, it provides limited supervision on rendered motion quality. To close this perception-action loop, we propose a *Rendering-Aware Reinforcement Learning* strategy using Group Relative Policy Optimization (GRPO) (Liu et al., 2024). By incorporating feedback from a video perception encoder (PE-Core), our approach directly optimizes semantic alignment, enabling complex non-rigid deformations beyond the reach of code-only supervision.

To address the data bottleneck, we introduce **SVGAnim-134k**, the first large-scale, high-quality dataset for vector animation. Unlike prior sketch-based datasets, SVGAnim-134k consists of professionally authored animations featuring closed shapes, rich semantics, and complex hierarchical grouping. All samples are processed through a rigorous topological canonicalization pipeline to ensure consistency and suitability for learning sparse state transitions.

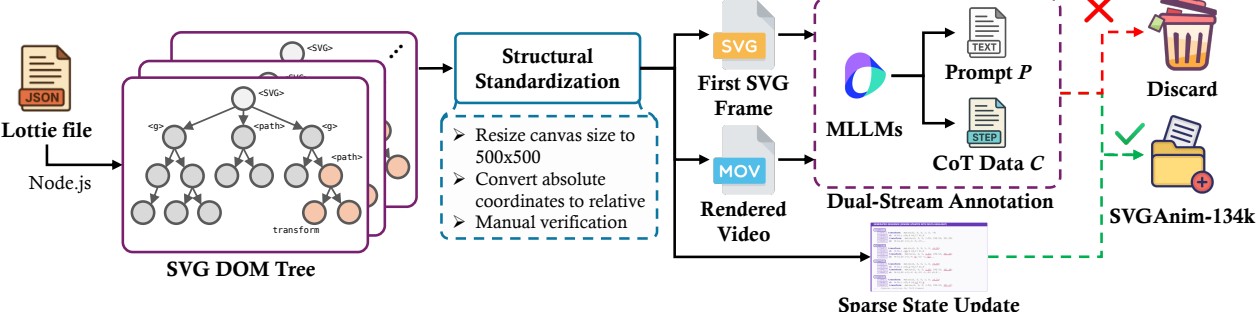

*Figure 2.* **Construction Pipeline of SVGAnim-134k.** The process operates in three stages: (1) **Structural Standardization:** Raw Lottie files are rendered into ID-anchored SVG DOM trees and normalized to a unified coordinate space. (2) **Sparse Encoding:** The animation sequence is compressed into *Sparse State Updates*, represented by serialized differential tokens (visualized in purple). (3) **Dual-Stream Annotation:** An MLLM generates both User Prompts ($P$) and Structure-Bound CoT ($C$). Samples failing the ID-consistency check are automatically discarded to ensure high-fidelity grounding.

In summary, our contributions are threefold:

- **Large-Scale Vector Animation Dataset**: We introduce SVGAnim-134k and establish the first benchmark for training and evaluating generative SVG animation models.

- **VAnim Framework**: We propose a novel generative paradigm that combines Identification-First Motion Planning with Sparse State Updates, enabling efficient long-horizon generation while preserving the original SVG structure by construction.

- **Rendering-Aware Optimization**: We demonstrate that GRPO with visual feedback substantially improves motion quality, allowing LLMs to learn geometry-level deformations beyond affine transformations.

## 2. SVGAnim-134k: A Large-Scale Benchmark for Vector Animation

Training a model to comprehend the interplay between static geometry and temporal dynamics requires data that is both large-scale and structurally rigorous. Existing vector datasets (Lopes et al., 2019; Carlier et al., 2020) focus exclusively on static images, while animation datasets are predominantly raster-based. To bridge this gap, we introduce **SVGAnim-134k**, the first large-scale dataset comprising 134,000 high-quality, linguistically annotated vector animation sequences.

This section details our data construction pipeline, as illustrated in Figure 2. The process encompasses data acquisition, topological canonicalization, sparse state update extraction, and dual-stream annotation generation.

### 2.1. Data Acquisition and Topological Canonicalization

We construct SVGAnim-134k by sourcing Lottie animation files (JSON format) (Lottie Community, 2024) from

Flaticon (Flaticon, 2026), a repository hosting professional-grade vector animations for UI and web design. This source provides structured, noise-free designs with complex grouping and fill rules.

**Data Structure Definition.** Before preprocessing, it is crucial to define the underlying representation. We treat each animation as a temporal sequence of topologically isomorphic SVG DOM trees. Unlike raster videos composed of pixel arrays, the dynamics in our dataset are encoded exclusively through the evolution of node attributes. These include: (1) *Geometry Attributes* (e.g., path data d), which define the shape's Bezier curves; (2) *Transformation Attributes* (e.g., transform), governing translation, rotation, and scaling; and (3) *Appearance Attributes* (e.g., fill-opacity, stroke), determining visual style. This explicit attribute-driven structure serves as the physical basis for our subsequent sparse state modeling.

To transform the parametric Lottie files into this explicit SVG sequence format, we utilize a Node.js-based rendering script (Airbnb, 2015). A critical advantage of this pipeline is that the generated SVGs inherently share an identical SVG DOM tree structure across frames. Leveraging this natural topological invariance, we index all tags involved in dynamic updates with globally unique identifiers that persist throughout the animation sequence.

Finally, we implement a rigorous structural standardization protocol to ensure learnability: (1) normalizing the viewport to a $500 \times 500$ px resolution, which matches the native resolution of most assets while balancing visual fidelity and token efficiency; (2) converting absolute coordinates to relative ones to compress sequence length; and (3) performing manual verification to filter out flawed rendering samples.

### 2.2. Sparse State Update Extraction

Standard autoregressive approaches represent animation as a sequence of full SVG codes, i.e., $(S_0, S_1, \ldots, S_T)$.

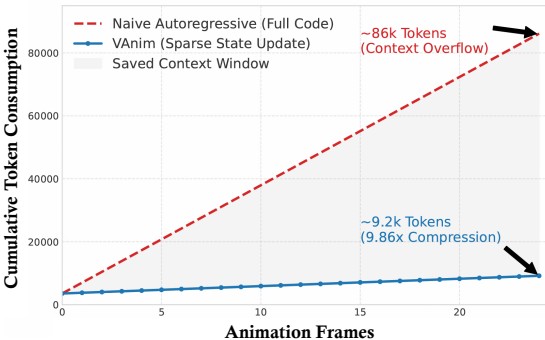

*Figure 3.* **Token Efficiency Analysis.** A comparison of cumulative token consumption over a 24-frame sequence. Naive frame-by-frame generation (red dashed line) suffers from *context explosion*, reaching ∼86k tokens. In contrast, VAnim's *Sparse State Update* mechanism (blue solid line) maintains a compact representation (∼9.2k tokens), achieving a **9.86× compression ratio**. This efficiency is critical for enabling long-horizon generation within limited context windows.

However, we observe that more than $85\%$ of the syntax (e.g., complex path data `d`, style definitions) remains unchanged between adjacent frames. Repeating this redundant information wastes computational resources and increases the risk of *identity drift*, where static elements are unintentionally modified during regeneration.

To address this issue, we reformulate the representation as a **Sparse State Update (SSU)** sequence. Let $A(S_t)$ denote the set of attributes at frame $t$. We define the sparse state update $\Delta_t$ as the set of attribute differentials relative to the previous frame:

$$\Delta_t = \{(id, attr, v_t) \mid v_t \neq v_{t-1}, \ (id, attr, v_t) \in A(S_t)\},$$

where $v_t$ denotes the value of attribute $attr$ for node $id$ at frame $t$. The complete animation is thus represented as $(S_0, \Delta_1, \ldots, \Delta_T)$

**Serialized Representation.** We visualize a concrete segment of this extracted sequence in Figure S2. As illustrated, the hierarchical updates are serialized into a token stream using special control tags (`<|time=t|>`, `<|ID=id|>`). This format explicitly anchors dynamic changes to persistent SVG DOM nodes, ensuring that only the evolving parameters (highlighted with underlines in the figure) are generated.

**Token Efficiency Analysis.** As shown in Fig. 3, this sparse formulation fundamentally alters the learning landscape. Representing a 24-frame animation using full SVG code requires an average of 86.0k tokens. In contrast, our $(S_0, \Delta_{1:T})$ representation reduces this to 9.2k tokens, achieving a $9.86\times$ compression ratio. Specifically, we restrict ID indexing to potentially dynamic nodes (e.g., `path`, `rect`, `g`) and filters (e.g., `feColorMatrix`, `feMorphology`), ignoring static metadata. Consequently,

the diff component constitutes **61%** of the total sequence length. Unlike naive generation where the vast majority of tokens repeat static redundancy, this high proportion indicates that the model's representational capacity is efficiently allocated to modeling dynamic state transitions.

### 2.3. Dual-Stream Annotation Pipeline

To enable the model to translate abstract user intent into concrete structural updates, we develop an automated annotation pipeline that generates two complementary textual modalities. We leverage *Doubao-Seed-1.6* (ByteDance Seed Team, 2025), a state-of-the-art multimodal model, to analyze the initial SVG structure $S_0$ and the rendered visual dynamics $V$. By aligning code topology with visual phenomenology, we produce high-fidelity supervision for both the input prompts ($P$) and the intermediate reasoning ($C$).

**User-Centric Prompt Generation ($P$).** First, we prompt the VLM to simulate a human user describing the animation. The goal is to capture the *visual phenomenology*, what the animation looks like, rather than its implementation details. We encourage diverse linguistic styles, ranging from concise commands to descriptive requests. These instructions serve as the conditional input $P$ during training.

**Structure-Bound Chain-of-Thought ($C$).** Second, to bridge the gap between the user prompts $P$ and the sparse updates $\Delta$, we generate a structured reasoning chain. To prevent hallucinations, we constrain this Structure-Bound CoT to a strict two-stage format:

- *Entity Identification:* Explicitly maps visual elements to persistent SVG IDs (e.g., "*The blue circle corresponds to ID `05`*").

- *Visual Dynamic Planning:* Describes the temporal logic grounded in these IDs (e.g., "*ID `05` must scale up and down while maintaining its center position*").

**Strict ID-Consistency Filter.** To ensure the reliability of the Structure-Bound CoT supervision, we implement a rigorous verification mechanism. For every generated reasoning chain, we cross-reference all referenced entity IDs against the actual SVG DOM tree of $S_0$. If a chain cites a non-existent ID or misidentifies a node type, the sample is discarded. This guarantees that 100% of the training data is structurally grounded, providing a robust foundation for *Identification-First Motion Planning*.

### 2.4. Dataset Statistics and Splitting

SVGAnim-134k covers a diverse range of categories, including UI icons, loading indicators, and narrative illustrations. To support our two-stage training paradigm, we partition the dataset into three subsets with distinct roles. The

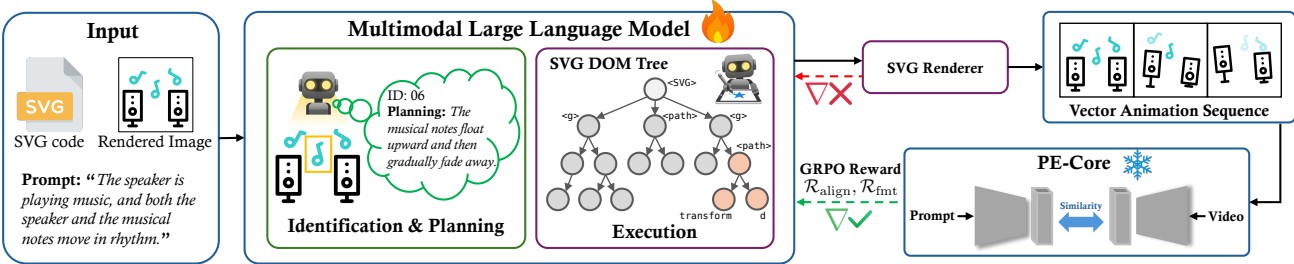

*Figure 4.* **Overview of the VAnim Framework.** The generation process is decomposed into two stages: (1) *Identification-First Motion Planning*, where the Multimodal LLM explicitly grounds visual entities to persistent SVG IDs (e.g., binding the object inside the orange box to `id=06` through reasoning) and reasons about causal motion logic; (2) *Sparse State Update*, where the model predicts attribute differentials ($\mathcal{D}_t$) only for the targeted nodes on the SVG DOM tree, guaranteeing topological isomorphism. Crucially, to bridge the *non-differentiable sampling and rendering gap*, we employ *Rendering-Aware Reinforcement Learning*. Using Group Relative Policy Optimization (GRPO), we align the discrete code updates with continuous visual dynamics via hybrid rewards from the PE-Core perception encoder.

**SVGAnim-SFT** subset contains 123k samples and serves as the primary corpus for Supervised Fine-Tuning (SFT), spanning the full breadth of animation semantics and equipping the model with foundational SVG literacy, robust ID anchoring, and syntactic correctness for sparse updates. The **SVGAnim-RL** subset consists of 10k high-complexity samples curated for Reinforcement Learning (RL), selected based on SVG geometric complexity to emphasize dense path structures and pronounced path-level deformations. This design encourages the model to focus on learning non-linear geometric transformations encoded in the SVG path `d` attribute, i.e., Bézier curve manipulation, thereby concentrating the computationally expensive RL stage on complex non-rigid dynamics rather than basic syntax or rigid motion primitives. The remaining 1k samples are reserved as a held-out test set for final evaluation.

## 3. Methodology

### 3.1. Problem Formulation

Given an initial static SVG $S_0$, its rendered raster image $I_0$, and a natural language instruction $P$, our goal is to generate a sequence of sparse state updates: $\mathcal{D} = \{\Delta_t \mid t = 1, \ldots, T\}$, where $T$ denotes the animation frames number.

To bridge the gap between abstract semantic instructions and low-level code execution, we introduce a latent *Structure-Bound Chain-of-Thought* (CoT) variable $C$ (Wei et al., 2022). We decompose the joint distribution into two stages:

$$p_\theta(o \mid x) = \underbrace{p_\theta(C \mid x)}_{\text{Planning Stage}} \cdot \underbrace{p_\theta(\mathcal{D} \mid C, x)}_{\text{Execution Stage}}$$

Here, $x = (I_0, S_0, P)$ denotes the input consisting of the initial SVG structure, visual context, and prompt, while $o = (C, \mathcal{D})$ represents the planned reasoning trace and the corresponding sparse state updates.

### 3.2. Architecture

VAnim is instantiated as a Multimodal Large Language Model (MLLM) $\pi_\theta$ (Bai et al., 2025). The architecture consists of two primary components: (1) **A Visual Encoder** $\mathcal{E}_v$, which maps the rendered raster image $I_0$ into a sequence of continuous visual embeddings $h_v$. (2) **A LLM Decoder**, which processes the discrete textual sequence formed by the SVG code $S_0$ and the user prompts $P$. Critically, the visual embeddings $h_v$ are projected into the LLM's token space and interleaved with the text embeddings. This unified representation enables the model to perform dense cross-modal attention, aligning the visual appearance of objects in $I_0$ with their corresponding SVG DOM nodes (IDs) in $S_0$.

### 3.3. Inference Process

Unlike conventional autoregressive decoding that directly maps text to code, inference in VAnim is explicitly structured as a coarse-to-fine, role-based process. This design decouples high-level semantic grounding from low-level geometric execution, enabling precise control while preserving object identity and topology.

**Coarse-to-Fine Inference Paradigm.** Given the input $(I_0, S_0, P)$, inference proceeds in two stages. The model first performs global semantic reasoning to identify relevant entities and intended motions, and then executes localized attribute updates constrained to the SVG DOM structure. This separation is critical for preventing identity drift and unbounded structural modification during generation.

**Identification-First Motion Planning.** In the first stage, the model $\pi_\theta$ acts as a *Director*, encoding the full multimodal context and synthesizes an explicit reasoning chain $C$. As described in Section 2.2, this chain grounds textual instructions to specific SVG entity IDs and attributes, producing a structured motion blueprint. By explicitly resolving *what* to move and *where*, this planning step sharply restricts the

solution space for subsequent generation.

**Sparse State Update Execution.** Conditioned on the motion blueprint $C$, the model then acts as an *Animator* to generate a sequence of sparse state updates $\mathcal{D}$. Rather than rewriting the full SVG code, the model predicts token-level differentials within the sparse update format. This execution mechanism constrains updates to the initial SVG DOM topology, preserving identity for non-participating elements and maintaining structural consistency throughout the animation.

### 3.4. Stage I: Structured Supervised Fine-Tuning

To initialize the parameters $\theta$, we first train the model to master *SVG literacy*: understanding SVG DOM hierarchies, following the Structure-Bound CoT format, and producing valid YAML-based diffs.

We use the **SVGAnim-SFT** subset for this cold-start stage. The objective maximizes the likelihood of the ground-truth planning chain $C$ and sparse updates $\mathcal{D}$:

$$\mathcal{L}_{\text{SFT}}(\theta) = -\mathbb{E}_{(I_0, S_0, P) \sim D_{\text{SFT}}} \Big[ \log p_\theta(C, \mathcal{D} \mid I_0, S_0, P) \Big]$$

### 3.5. Stage II: Rendering-Aware Reinforcement Learning

While SFT equips the model with syntactic correctness and basic logical structure, it remains blind to rendered visual dynamics and perceptual motion quality. To bridge the gap between discrete code generation and continuous visual perception, we introduce a Rendering-Aware Reinforcement Learning (RL) stage trained on the **SVGAnim-RL** subset.

For each input $x = (I_0, S_0, P)$, we sample a group of $G$ candidate outputs $\{o_1, \ldots, o_G\}$, where each $o_i = (C_i, \mathcal{D}_i)$ represents a set of sparse state updates. Each candidate is rendered into a video sequence and evaluated using a hybrid reward function based on both visual semantics and structural validity.

We optimize the policy using Group Relative Policy Optimization (GRPO) (Shao et al., 2024; Schulman et al., 2017), with the following objective:

$$\mathcal{L}_{\text{GRPO}}(\theta) = \mathbb{E}_{x \sim D_{\text{RL}}} \left[ \frac{1}{G} \sum_{i=1}^{G} \min \left( \frac{\pi_\theta(o_i \mid x)}{\pi_{\theta_{\text{old}}}(o_i \mid x)} \hat{A}_i, \text{clip}(\cdot) \hat{A}_i \right) - \beta D_{\text{KL}} \right]$$

This formulation encourages exploration of the solution space beyond conservative rigid transformations, enabling fine-grained manipulation of Bézier control points.

**Hybrid Reward Mechanism.** To align sparse state updates with user intent while ensuring executability, we design a *dual-objective reward function* that jointly optimizes *semantic fidelity* and *format validity*.

**Semantic Alignment Reward.** To capture fine-grained motion semantics described in the prompt, we leverage PE-

Core (Bolya et al., 2025), a SOTA video-text perception encoder. We compute the cosine similarity between the textual instruction $P$ and the rendered animation $V_{\text{pred}}$:

$$\mathcal{R}_{\text{align}} = \text{CosineSim}(\text{E}_{\text{text}}(P), \text{E}_{\text{video}}(V_{\text{pred}}))$$

Unlike SFT, this reward provides a direct visual learning signal. Maximizing $R_{\text{align}}$ incentivizes the model to manipulate fine-grained path attributes (e.g., the $\mathtt{d}$ attribute) rather than relying on restricted affine transforms, thereby enabling the precise execution of complex non-rigid behaviors such as elastic bending and fluid morphing.

**Format Validity Reward.** To ensure robust execution, we impose a strict binary validity reward. A generated sequence $\mathcal{D}$ receives a positive reward only if it satisfies three conditions: (1) it is syntactically parseable and renderable; (2) the sequence length exactly matches the target frame count $T$; and (3) all updates reference valid IDs in $S_0$ without breaking the topology.

$$\mathcal{R}_{\text{fmt}} = \begin{cases} 1, & \text{if } \mathcal{D} \text{ is valid, } |\mathcal{D}| = T, \text{ and structure is intact} \\ -1, & \text{otherwise} \end{cases}$$

This signal explicitly penalizes syntax errors, length mismatches, and ID hallucinations, effectively confining policy exploration to the executable and topologically consistent manifold. The final reward is defined as:

$$\mathcal{R} = \lambda_{\text{align}} \mathcal{R}_{\text{align}} + \lambda_{\text{fmt}} \mathcal{R}_{\text{fmt}}$$

## 4. Experiments

### 4.1. Implementation Details

**Model Architecture and Training.** VAnim is built upon the Qwen3-VL-8B-Thinking backbone (Bai et al., 2025). We employ a two-stage training pipeline on $8\times$ NVIDIA H100 GPUs. In Stage I, we perform full-parameter structured supervised fine-tuning on the SVGAnim-SFT dataset. The maximum sequence length is set to 25k tokens. In Stage II, we apply GRPO with a group size of $G = 8$, sampling temperature $T = 0.9$, and KL coefficient $\beta = 0.01$. To provide high-fidelity feedback, animations are rendered via a headless Playwright browser (Microsoft, 2026) at $500 \times 500$ resolution, ensuring the capture of subtle non-rigid deformations. The overall reward function adopts an equal-weighting strategy with $\lambda_{\text{align}} = 1.0$ and $\lambda_{\text{fmt}} = 1.0$.

**Baselines and Evaluation.** For baseline comparisons, we evaluate proprietary LLM and optimization-based methods. For GPT-5.2 and Gemini 3 Pro, we employ a multimodal prompting strategy, where each prompt includes both the initial SVG code and its rendered image, ensuring maximal contextual grounding. For the optimization-based LiveSketch method, we use the official implementation and increase the optimization budget to 1,000 SDS steps to guar-

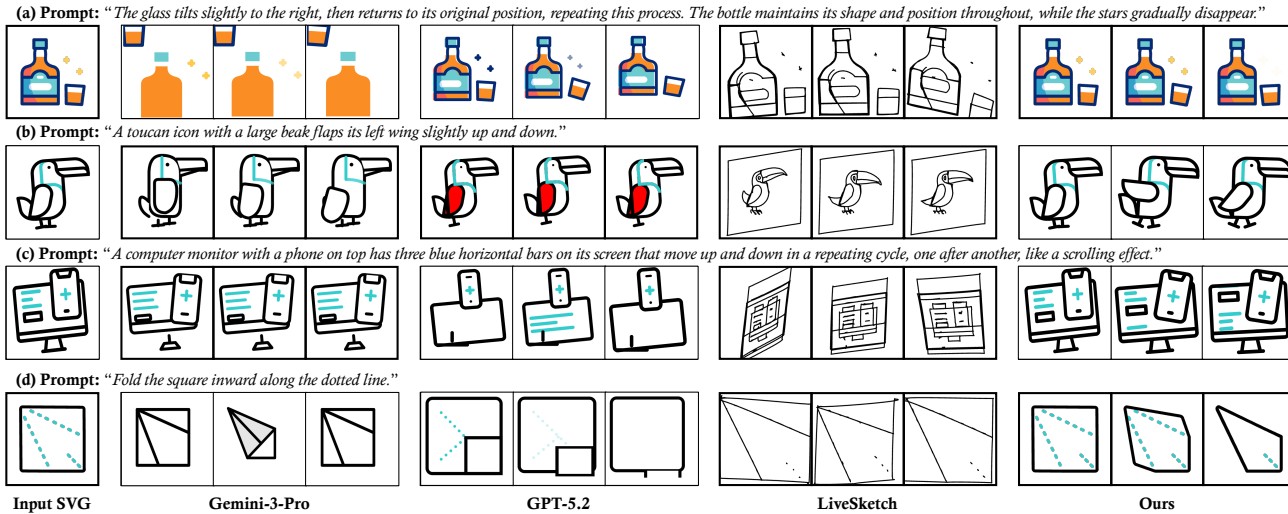

**(a) Prompt:** *"The glass tilts slightly to the right, then returns to its original position, repeating this process. The bottle maintains its shape and position throughout, while the stars gradually disappear."*

**(b) Prompt:** *"A toucan icon with a large beak flaps its left wing slightly up and down."*

**(c) Prompt:** *"A computer monitor with a phone on top has three blue horizontal bars on its screen that move up and down in a repeating cycle, one after another, like a scrolling effect."*

**(d) Prompt:** *"Fold the square inward along the dotted line."*

| Input SVG | Gemini-3-Pro | GPT-5.2 | LiveSketch | Ours |

*Figure 5.* **Qualitative Comparison on SVGAnim-Test.** We compare VAnim against state-of-the-art baselines on four challenging prompts requiring precise control and non-rigid deformation. Baselines suffer from severe Identity Drift and Topological Collapse. Specifically, GPT-5.2 changes the toucan's wing color to red; Gemini distorts the bottle shape; while LiveSketch produces broken lines. Our **VAnim** generates smooth, semantically aligned animations while strictly preserving the visual identity and topology of the original vector graphics, thanks to its ID-anchored sparse update mechanism.

antee convergence and eliminate under-optimization as a confounding factor.

For evaluation, we measure semantic alignment using the *PE-Core-G14-448* video perception encoder (Bolya et al., 2025). Following the encoder's native protocol, all generated videos are resized to $448 \times 448$ via bicubic interpolation prior to feature extraction, ensuring stable and reliable metric computation. To reduce metric circularity, the supplementary material additionally reports an independent evaluation suite with InternVideo2, mean flow magnitude, flow_tLPIPS, and SSIM.

### 4.2. Quantitative Comparison

We evaluate performance using two complementary metrics: *Semantic Alignment*, measured by PE-Core cosine similarity between the prompt and the rendered animation, and *Success Rate*, defined as the percentage of syntactically valid and renderable outputs. We present the quantitative evaluation on the SVGAnim-Test set in Table 1. VAnim-GRPO demonstrates superior performance across both metrics, achieving a state-of-the-art Semantic Alignment of 0.281 and a perfect Success Rate of 100%. The appendix extends this comparison with an independent video-text metric (InternVideo2) as well as temporal smoothness (flow_tLPIPS), motion magnitude, and identity preservation (SSIM).

**Limitations of Baselines.** The quantitative gap highlights the structural flaws in existing paradigms. First, the low semantic score of LiveSketch (0.158) exposes its inherent *topological instability*. By treating vector graphics as col-

*Table 1.* **Quantitative Comparison on SVGAnim-Test.** VAnim-GRPO achieves the highest semantic alignment while guaranteeing perfect executability.

| Method | Semantic Alignment ↑ | Success Rate ↑ |
|---|---|---|
| LiveSketch | 0.158 | **100.0%** |
| GPT-5.2 | 0.234 | 88.5% |
| Gemini 3 Pro | 0.243 | 86.2% |
| VAnim (SFT-only) | 0.268 | 95.2% |
| **VAnim (GRPO)** | **0.281** | **100.0%** |

lections of independent strokes, it lacks semantic awareness of closed shapes and hierarchical groupings, causing filled regions to merge or collapse under occlusion. Second, while LLMs like GPT-5.2 possess strong reasoning, they suffer from *domain misalignment* and *syntax degradation*. They tend to default to generic rigid motions (e.g., simple translation) rather than executing non-rigid deformations. More critically, their free-form generation often succumbs to unclosed tags or hallucinated attributes in long-horizon tasks, rendering the output invalid.

**Sources of VAnim's Superiority.** In contrast, VAnim's performance stems from two core methodological innovations. (1) *Rendering-Aware RL:* The improvement from SFT (0.268) to GRPO (0.281) confirms that the visual reward signal effectively bridges the gap between code and semantics, incentivizing the model to manipulate fine-grained path attributes (d) to achieve higher fidelity. The appendix further shows that removing either the alignment reward or the format reward degrades semantic quality or executability,

*Table 2.* **Ablation Study.** We evaluate the impact of Rendering-Aware RL and Structure-Bound CoT.

| Variant | Semantic Alignment ↑ | Success Rate ↑ |
|---------|---------------------|----------------|
| **Full VAnim (Ours)** | **0.281** | **100.0%** |
| w/o Rendering-Aware RL | 0.268 (-0.013) | 95.2% (-4.8%) |
| w/o Structure-Bound CoT | 0.255 (-0.026) | 98.6% (-1.4%) |

respectively. (2) *Sparse Architecture:* By formulating generation as Sparse State Updates, we inherently constrain the output space to valid attribute differentials. The appendix additionally shows that a naive frame-by-frame variant without SSU collapses to 62.3% success rate, while the input-image ablation degrades SSIM and temporal smoothness, confirming that structure-aware representation and visual grounding are both critical.

### 4.3. Qualitative Analysis

We present representative results in Figure 1 and Figure 5, covering object interaction, multi-part coordination, and non-rigid deformation.

**Identity locking and topology preservation.** VAnim preserves the input SVG's visual identity (e.g., stroke width, color, and component geometry) while producing motion, enabled by ID-anchored Sparse State Updates. In contrast, LLMs may introduce structural drift or hallucinated elements (e.g., unexpected color/part changes), and optimization-based methods can break closed shapes or fill regions under occlusion.

**Geometry-level deformation.** VAnim performs genuine path-level editing by directly manipulating $d$ to realize non-rigid effects such as folding or morphing. Baseline systems that rely on CSS/SMIL affine transforms typically fail to express such geometry changes, often degenerating to rigid motion or visually inconsistent deformation.

**Fine-grained instruction following.** VAnim reliably executes nuanced temporal logic (e.g., sequential scrolling) while keeping irrelevant attributes unchanged. By comparison, LLMs often default to generic rigid motion, and optimization-based approaches may suffer from artifacts that reduce semantic faithfulness.

### 4.4. Ablation Study

To validate the contribution of our core components, we conduct quantitative (Table 2) and qualitative (Figure 6) comparisons against two variants: (1) *w/o Rendering-Aware RL* (SFT baseline) and (2) *w/o Structure-Bound CoT*. The appendix expands this analysis with a no-SSU baseline, a no-input-image variant, reward ablations, and a GRPO group-size sweep.

**Impact of Rendering-Aware RL** Removing the RL stage

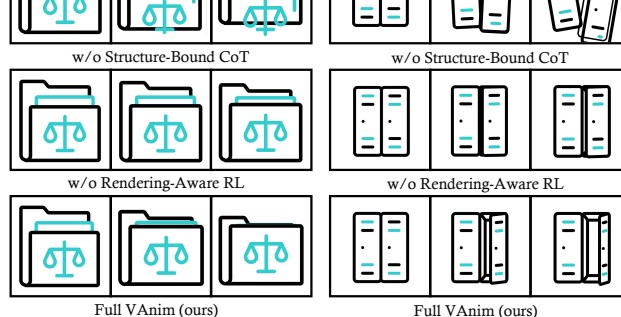

(a) **Prompt:** "*A folder with a scales-of-justice motif gradually closes: the outer layer moves upward, while the inner layer moves downward.*" (b) **Prompt:** "*The door of the room on the right gradually opens fully, while the room on the left remains still.*"

*Figure 6.* **Qualitative Ablation.** Row 1 (*w/o Structure-Bound CoT*) suffers from **grounding failures**, manipulating wrong entities (e.g., rotating the whole cabinet instead of the door). Row 2 (w/o RL) exhibits **motion laziness**, producing conservative updates that fail to fully satisfy the "close" or "open" instructions. Row 3 (Full) achieves accurate semantic alignment with precise deformation.

causes a semantic drop (0.281 → 0.268). Qualitatively, the SFT model exhibits **"conservative motion bias"**. As shown in Figure 6 (Row 2), the folder fails to close completely, and the door only opens a slight crack, failing to satisfy the prompts "opens fully". This confirms that the visual reward signal is essential to push the model beyond lazy, minimal updates, ensuring the *magnitude* of deformation aligns with the textual intensity.

**Impact of Structure-Bound CoT** Removing the planning stage results in the most significant degradation (0.281 → 0.255). Without explicit entity identification, the model fails to map instructions to the correct SVG DOM nodes. In Figure 6 (Row 1), instead of opening the door component, the model incorrectly rotates the entire cabinet structure (b) or detaches the scale icon (a). This demonstrates that structured reasoning is a prerequisite for **structural integrity**, preventing the model from manipulating the wrong entities or breaking topological layouts.

**Additional Appendix Ablations.** Beyond the main ablations above, the supplementary experiments show that the no-SSU baseline drops success rate to 62.3% and sharply worsens temporal smoothness, the no-input-image setting reduces SSIM and degrades motion quality, and the GRPO hyperparameter sweep (G=4/8/16) exposes a trade-off between semantic alignment and identity preservation.

### 4.5. User Study

We further conduct a user study to evaluate perceptual quality beyond automatic metrics. As summarized in Table S2, VAnim significantly outperforms all baselines across vi-

sual integrity, motion smoothness, and instruction following. More details are provided in the supplementary material.

## 5. Conclusion

In this work, we introduced **VAnim**, the first LLM-based framework enabling open-domain text-to-SVG animation generation. Crucially, we addressed the long-standing data scarcity bottleneck by constructing **SVGAnim-134k**, the first large-scale, high-quality dataset for vector animation.

Building upon this data, we identify that context explosion and identity drift arise from a representation mismatch. VAnim resolves this by reformulating animation as *Sparse State Updates* on a persistent SVG DOM tree. This paradigm, coupled with *Identification-First Motion Planning*, preserves the SVG DOM structure and non-participating elements by construction while improving identity consistency for animated content. Furthermore, our *Rendering-Aware Reinforcement Learning* strategy helps close the loop between discrete code and continuous visual dynamics.

Extensive experiments verify that VAnim not only achieves state-of-the-art semantic alignment and structural validity but also establishes a new benchmark for executing complex, path-level non-rigid deformations in open-domain vector animation. The supplementary evaluation further reports independent metrics for temporal smoothness, identity preservation, and motion magnitude. We hope that VAnim and the open-sourced SVGAnim-134k will serve as a strong baseline, inspiring future research into intelligent, structure-preserving vector content creation.

**Future Work.** While VAnim excels at visual animation, extending this paradigm to model interactive behaviors (e.g., generating JavaScript triggers) and handling longer, multi-scene narratives remain promising frontiers.

## Acknowledgements

This work was supported in part by in part by National Natural Science Foundation of China (No.62572039, No.62461160331, No.62132001) and Young Elite Scientists Sponsorship Program by CAST. This work was also supported by the NSFC/RGC Collaborative Research Scheme (CRS_HKU703/24). Dr. Xu's research work described in this paper was conducted in the JC STEM Lab of Multimedia and Machine Learning funded by The Hong Kong Jockey Club Charities Trust.

## Impact Statement

SVG animation generation has significant potential for positive applications in professional design, UI/UX development, content creation, and accessibility. However, like other generative technologies, VAnim could potentially be misused to create deceptive or manipulated content. We advocate for responsible deployment and emphasize the intended use of VAnim as a creative assistant for professional designers and developers.

The method has computational overhead from rendering-aware RL, which may limit accessibility for resource-constrained users. The approach is primarily designed for Lottie-style vector graphics, and generalization to hand-authored or tool-exported SVGs remains an open question. We encourage transparency in AI-generated media and support the development of detection mechanisms to prevent misuse.

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

## Technical Appendices and Supplementary Material

## Overview

This supplementary material is organized into several sections that provide additional details and analysis related to our work on VAnim. Specifically, it includes the following aspects:

- In Section A, we present the full evaluation table with additional baselines, input/reward ablations, and GRPO sensitivity analysis.

- In Section B, we provide detailed implementation information for VAnim.

- In Section C, we describe a user study that demonstrates the superiority of our method compared to existing approaches.

- In Section D, we present additional qualitative results generated by VAnim.

- In Section E, we introduce the related work of VAnim.

- In Section F, we discuss limitations and the potential societal impact of VAnim.

## A. Comprehensive Evaluation, Baselines, and Ablations

Table S1 reports the expanded evaluation matrix referenced in the rebuttal. Compared with the compact main-table view, this version adds an independent semantic metric (InternVideo2), motion magnitude, temporal smoothness, and identity preservation. It also surfaces the practical ablations that isolate the contribution of Sparse State Updates, visual grounding, and each reward term.

### A.1. Expanded Analysis

The expanded table confirms the three main takeaways from the rebuttal. First, the independent InternVideo2 score remains consistent with the main conclusions, which reduces reliance on PE-Core alone. Second, the no-SSU baseline is the most revealing control: frame-by-frame regeneration preserves some visual quality on easy cases but collapses to 62.3% success rate and much worse temporal smoothness. Third, the reward and input ablations separate the roles of the visual encoder, semantic reward, and format reward: removing $R_{align}$ lowers semantic fidelity and motion magnitude, removing $R_{fmt}$ mainly hurts executability, and removing the input image degrades identity preservation and motion consistency.

The GRPO sweep further shows that larger group sizes improve semantic alignment and motion magnitude at the cost of slightly reduced SSIM. We therefore keep $G = 8$ as the default trade-off between exploration and structural stability.

## B. Implementation Details of VAnim

This section provides practical implementation details of the VAnim framework, including supervised fine-tuning (SFT), rendering-aware reinforcement learning (RLRF), and engineering strategies for stable training.

### B.1. Supervised Fine-Tuning (SFT)

We perform structured supervised fine-tuning using the **LLaMA-Factory** codebase (Zheng et al., 2024). The base model is **Qwen3-VL-8B** (Bai et al., 2025), initialized from an instruction-tuned checkpoint. SFT serves as a cold-start stage, teaching the model SVG syntax, DOM awareness, structured Chain-of-Thought formatting, and sparse YAML-based diff prediction (Consortium, 1999; W3C, 2020; SVG Working Group, 2025; MDN contributors, 2025; W3C, 2019).

**Training Configuration.** We adopt full-parameter fine-tuning with FlashAttention-2 enabled (Dao, 2023). The maximum sequence length is set to **25k tokens** to accommodate complex SVG structures and planning traces. Due to memory constraints, we use a micro-batch size of 1 with gradient accumulation over **32 steps**. We use AdamW (Loshchilov & Hutter, 2019) with a learning rate of $1 \times 10^{-4}$ and weight decay 0, and adopt a cosine scheduler with 5% warmup. Training is performed in bf16 precision for **2** epochs.

We employ DeepSpeed ZeRO-3 (Rajbhandari et al., 2020) for full parameter sharding across 8 GPUs. Evaluation is conducted every 500 steps on a held-out validation split. The resulting checkpoint provides a strong initialization for subsequent reinforcement learning.

*Table S1.* **Comprehensive Evaluation, Baselines, and Ablation Studies.** We introduce a multi-dimensional evaluation matrix utilizing an independent multimodal model (InternVideo2) to reduce metric circularity, alongside metrics for motion magnitude (mean flow mag), temporal smoothness (flow tLPIPS), and identity preservation (SSIM). **SR** denotes the success rate of generating executable and topologically valid SVGs. Best baseline results are in **bold**.

| Method / Variant | InternVideo2 (↑) *Semantic Align* | mean_flow_mag *Motion Mag* | flow_tLPIPS (↓) *Temporal Smooth* | SSIM (↑) *Identity Preserv* | SR (↑) *Executability* |
|---|---|---|---|---|---|
| ***Main Results & Baselines*** | | | | | |
| AniClipart (Wu et al., 2024) | 0.092 | 0.927 | 0.0376 | 0.9278 | 100% |
| FlipSketch (Bandyopadhyay & Song, 2025) | 0.137 | 1.696 | 0.1575 | 0.6786 | 100% |
| GPT-5.2 | 0.180 | 0.954 | 0.0148 | 0.9505 | 88.5% |
| Gemini 3 Pro | 0.182 | 0.804 | 0.0136 | 0.9634 | 86.2% |
| LiveSketch (Gal et al., 2024) | 0.107 | 0.801 | 0.0612 | 0.9000 | 100% |
| **VAnim (Ours)** | **0.202** | **1.711** | **0.0117** | **0.9719** | **100%** |
| ***Ablations on Architecture & Inputs*** | | | | | |
| NO SSU (Naive Frame-by-Frame) | 0.161 | 1.412 | 0.2070 | 0.9448 | 62.3% |
| No CoT (No Planning) | 0.172 | 1.745 | 0.1580 | 0.9514 | 98.6% |
| No Input Image (Text + Code Only) | 0.176 | 1.548 | 0.1650 | 0.9245 | 96.3% |
| ***Ablations on RL & Reward Mechanisms*** | | | | | |
| NO $R_{align}$ Reward | 0.191 | 1.589 | 0.1330 | 0.9811 | 100% |
| NO $R_{fmt}$ Reward | 0.199 | 1.705 | 0.1020 | 0.9734 | 96.6% |
| NO GRPO (SFT-only) | 0.187 | 1.512 | 0.1360 | 0.9756 | 95.2% |
| ***GRPO Hyperparameter (Group Size $G$)*** | | | | | |
| $G = 4$ | 0.195 | 1.598 | 0.0128 | 0.9801 | 100% |
| $G = 8$ *(VAnim Default)* | 0.202 | 1.711 | 0.0117 | 0.9719 | 100% |
| $G = 16$ | 0.207 | 1.743 | 0.0112 | 0.9689 | 100% |

## B.2. Rendering-Aware Reinforcement Learning via GRPO

To bridge the gap between discrete code generation and continuous visual dynamics, we further optimize the model using **Group Relative Policy Optimization (GRPO)** implemented with the **TRL** library (Shao et al., 2024; Hugging Face, 2026).

**Training Setup.** GRPO training is initialized from the SFT checkpoint and conducted for **2 epochs** over the SVGAnim-RL subset. Rollouts are generated autoregressively using the same Qwen3-VL model, with FlashAttention-2 enabled for efficiency. We use a batch size of 1 and accumulate gradients over **16 steps**. For each prompt, we sample $G = 8$ candidate generations per update. We use a learning rate of $2 \times 10^{-6}$ with AdamW ($\beta = (0.9, 0.99)$) and set the KL coefficient to $\beta = 0.01$, with weight decay 0.1. We adopt a cosine scheduler with 10% warmup, and set the maximum prompt and completion lengths to 12k and 18k tokens respectively.

We disable column pruning to preserve multimodal prompt structures and use bf16 precision throughout training.

## B.3. Perceptual Reward Computation

The core of rendering-aware reinforcement learning lies in perceptual reward computation. For each generated sparse update, we reconstruct the full SVG sequence and render it into a short video clip.

**SVG-to-Video Rendering.** We employ a custom SVG animation decoder based on a headless browser. To reduce overhead, a single global browser instance is reused across all rollouts (Microsoft, 2026). Each animation is rendered into a fixed-resolution video with up to 24 frames.

**Perceptual Encoding.** Rendered videos are uniformly sampled and encoded using the **PE-Core-G14** (Bolya et al., 2025; Meta AI, 2025) perception encoder. All embeddings are L2-normalized. Ground-truth video embeddings and text embeddings are precomputed and cached for efficiency. To avoid redundant computation, video embeddings are computed once per completion and reused across reward terms.

## C. User Study

Automatic metrics, while scalable, may not fully capture perceptual aspects such as topological integrity and motion realism. To rigorously evaluate the human-perceived quality of VAnim, we conduct a blind user study.

**Setup.** We recruited 15 participants, including professional UI designers and computer science researchers. Each participant evaluated 15 randomly selected test cases. For each case, participants were shown the input SVG, the text instruction, and anonymized animation results generated by LiveSketch, GPT-5.2, Gemini 3 Pro, and VAnim. All methods were presented in random order (Gal et al., 2024; OpenAI, 2025; Google Cloud, 2025).

**Ethics and Consent.** All participants provided informed consent prior to the study. The study involved perceptual ratings only, with no collection of personally identifiable information beyond basic demographic categorization for analysis. The study was conducted in accordance with institutional guidelines for human subjects research.

Participants rated each method using a 5-point Likert scale (Likert, 1932)(1 = Very Poor, 5 = Excellent) along three dimensions:

- **Visual Integrity:** Whether the object preserves its shape, identity, and topological structure without collapsing or flickering.

- **Motion Smoothness:** Whether the animation exhibits smooth and natural motion without noticeable jitter.

- **Instruction Following:** Whether the animation correctly and faithfully executes the textual instruction.

*Table S2.* **User Study Results.** Mean scores on a 1–5 Likert scale. VAnim achieves the highest ratings, particularly in **Visual Integrity**, demonstrating its superiority in preserving topological structure compared to optimization and pixel-based baselines.

| Method | Visual Integrity ↑ | Motion Smoothness ↑ | Instruction Following ↑ |
|---|---|---|---|
| LiveSketch | 2.15 | 2.43 | 2.12 |
| GPT-5.2 | 3.35 | 3.76 | 3.55 |
| Gemini 3 Pro | 3.42 | 3.85 | 3.68 |
| **VAnim (Ours)** | **4.62** | **4.48** | **4.55** |

**Results and Analysis.** The quantitative results are summarized in Table S2. VAnim consistently outperforms all baselines across all three evaluation dimensions.

On *Visual Integrity*, VAnim achieves a mean score of **4.62**, substantially higher than LiveSketch (2.15) and Gemini 3 Pro (3.42). This indicates that human evaluators strongly penalize topological breakage introduced by optimization-based methods, as well as identity drift observed in proprietary LLMs. These results validate the effectiveness of our ID-anchored sparse update mechanism.

In terms of *Instruction Following*, VAnim scores **4.55**. Participants noted that while Gemini 3 Pro often simplifies complex instructions into basic rigid transformations, VAnim is able to correctly execute nuanced non-rigid deformations such as shape morphing.

## D. Dataset Analysis

In this section, we provide a deeper visual analysis of the SVGAnim-134k benchmark to demonstrate the data foundation of VAnim.

### D.1. Dataset Diversity and Complexity

The generalization capability of VAnim is fundamentally underpinned by the diversity of its training data. As visualized in Figure S1, SVGAnim-134k transcends the limitations of simple icon datasets by incorporating high-entropy designs across multiple distinct categories:

- **UI Components & Functional Icons:** The dataset includes a vast array of user interface elements, such as credit cards, toggle switches, loading bars, and digital devices. These samples provide critical supervision for learning precise, rigid state transitions and logical interactions (e.g., sliding, toggling).

- **Organic & Non-Rigid Entities:** A significant portion of the data features biological entities (e.g., birds, bears, whales) and natural phenomena (e.g., moon phases, weather icons). Unlike rigid mechanical parts, animating these figures necessitates **non-rigid deformations**, such as tail wagging or shape morphing, which pushes the model to manipulate path data ($d$) rather than just affine transforms.

- **Complex Topology & Grouping:** Many samples, such as laboratory equipment and medical instruments, involve intricate multi-part groupings and layered structures. Exposure to such data enables VAnim to learn fine-grained coordination,

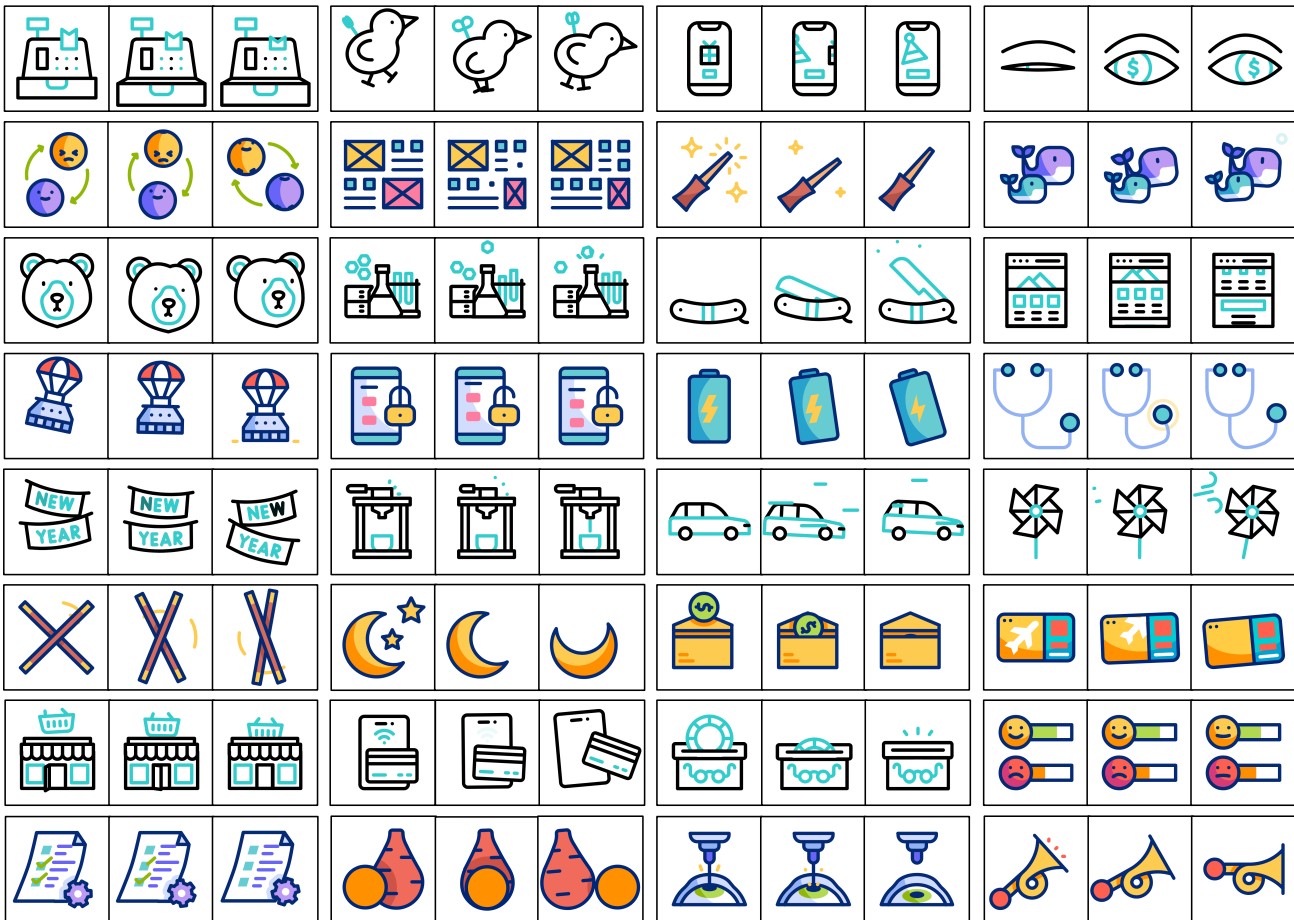

*Figure S1.* **Representative Samples from SVGAnim-134k.** Our dataset covers a diverse range of visual domains, including **UI Interaction** (e.g., loading bars, toggles), **Narrative Illustration** (e.g., magic wands, weather effects), and **Character Dynamics** (e.g., blinking eyes, walking animals). Unlike previous sketch-based datasets, these samples feature **professional-grade topology** with closed shapes, rich fill attributes, and complex occlusion layers, providing a rigorous foundation for learning structural vector animation.

ensuring that sub-components (e.g., the liquid inside a flask or the blades of a windmill) move in sync with their parent structures without topological breakage.

This rich semantic coverage ensures that VAnim does not merely overfit to a specific style but acquires a robust understanding of vector motion logic applicable to open-domain generation.

### D.2. Quantitative Analysis of Motion Primitives

To further quantify the motion complexity inherent in SVGAnim-134k, we conducted a statistical analysis of the attribute update types within the ground truth sequences (Original Data).

As illustrated in Figure S3 (Orange hatched bars), the dataset does not rely solely on simple affine transformations. While rigid motions (e.g., translation, rotation via `transform`) account for 70% of the state updates, a significant portion (30%) involves direct manipulation of path data (`d`). This distribution confirms that the dataset

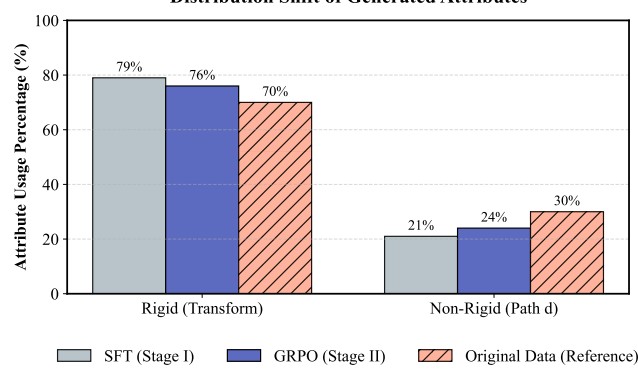

*Figure S3.* **Distribution of Motion Attributes.** We compare the usage frequency of rigid transformations (`transform`) versus non-rigid path deformations (`d`) across the training dataset (Original Data) and model generations. The ground truth data (Reference) exhibits a substantial proportion (30%) of path-level manipulations, confirming the complexity of the SVGAnim-134k benchmark.

```
GENERATED SEQUENCE (SPARSE UPDATES WITH DELTA HIGHLIGHT)

 <|time=1|>
    <|ID=2|>   transform:  matrix(1, 0, 0, 1, 0, -9)
    <|ID=20|>  d:  M-30,1 c16,6 44,17 61,9
    <|ID=21|>  transform:  matrix(1, 0, 0, 1.02, 292.53, 331.59)
    <|ID=23|>  d:  M-14,20 c-5,-6 -5,-15...
 <|time=2|>
    <|ID=2|>   transform:  matrix(1, 0, 0, 1, 0, -8.92)
    <|ID=20|>  d:  M-30,1 c15,6 44,17 61,9
    <|ID=21|>  transform:  matrix(1, 0, 0, 1.03, 292.53, 331.43)
    <|ID=23|>  d:  M-14,20 c-5,-6 -6,-15 -1,-22...
 <|time=3|>
    <|ID=2|>   transform:  matrix(1, 0, 0, 1, 0, -8.69)
    <|ID=20|>  d:  M-30,1 c15,7 44,17 61,9
    <|ID=21|>  transform:  matrix(1, 0, 0, 1.04, 292.53, 331.28)
    <|ID=23|>  d:  M-14,20 c-5,-6 -6,-15 -1,-22 c0,0...
 <|time=4|>
    <|ID=2|>   transform:  matrix(1, 0, 0, 1, 0, -8.33)
    <|ID=20|>  d:  M-30,1 c15,6 44,16 61,8
    <|ID=21|>  transform:  matrix(1, 0, 0, 1.04, 292.53, 331.17)
 ...   (Updates continue for T=24 frames)
```

*Figure S2.* **Structured Sparse Updates with Differential Highlighting.** We visualize the token sequence generated by VAnim. **Purple pills** (<|time=t|>) and **White pills** (<|ID=id|>) structure the hierarchy. Values marked with a **magenta underline** indicate the specific floating-point parameters that evolved from the previous frame. This highlights the model's ability to perform fine-grained, continuous modifications (e.g., path data morphing) rather than merely copying static attributes.

provides dense supervision for non-rigid deformations, such as shape morphing and organic movements.

Figure S3 also plots the attribute distributions of the generated animations from SFT and GRPO models for comparison. The alignment between the generated distributions and the original data serves as an indicator of how well the models capture the underlying motion priors of the dataset.

# E. Related Work

## E.1. Static Vector Graphics Generation and Editing

Prior work on static Scalable Vector Graphics (SVG) generation can be broadly categorized into (i) sequence-based command generation and (ii) optimization with differentiable rasterizers. Sequence modeling methods (e.g., DeepSVG (Carlier et al., 2020) and IconShop (Wu et al., 2023)) treat SVGs as drawing-command sequences, while optimization-based approaches (e.g., CLIPDraw (Frans et al., 2022), VectorFusion (Jain et al., 2023), and SVGDreamer (Xing et al., 2024)) refine vector primitives under vision-language objectives, typically via differentiable rendering (Li et al., 2020). Although effective for producing visually appealing graphics, optimization-driven methods often yield dense, weakly structured paths that hinder downstream editing and semantic part manipulation.

Recent Large Language Model (LLM) based methods further recast SVG creation as code synthesis, improving structural validity and editability (e.g., LLM4SVG (Xing et al., 2025), OmniSVG (Yang et al., 2025), SVGen (Wang et al., 2025b), Chat2SVG (Wu et al., 2025), and SVGThinker (Chen et al., 2025)). Meanwhile, RL-based refinement has been explored to optimize rendered perceptual quality for static SVGs (e.g., Reason-SVG (**?**) and rendering-aware RL (Rodriguez et al., 2025b)). However, these works primarily focus on static SVG generation or single-step edits, and do not address long-horizon temporal evolution with persistent identities.

## E.2. Vector Animation and Inbetweening

Classic computer graphics and HCI research has long studied vector/2D animation by enforcing correspondences, topological constraints, and artist controllability. For instance, topology-aware representations enable vector animations with changing structure (Dalstein et al., 2015), while correspondence-centric systems support semi-automatic inbetweening in production workflows (e.g., DiLight (Carvalho et al., 2017) and interactive tight inbetweening via BetweenIT (Whited et al., 2010)). More recently, learning-based approaches revisit stroke correspondence for 2D animation, such as JoSTC (Mo et al., 2024). These lines of work highlight that high-quality vector animation hinges on stable cross-frame identity/correspondence and topology/structure preservation, which are not directly addressed by modern text-to-SVG generators.

## E.3. Optimization-Based Vector Animation

Optimization-based vector animation methods typically deform sketches or clipart by iterative fitting under the supervision of pretrained generative models and differentiable rendering. LiveSketch (Gal et al., 2024) uses Score Distillation Sampling (SDS) (Poole et al., 2023) with text-to-video diffusion guidance, and follow-ups such as FlipSketch (Bandyopadhyay & Song, 2025) and AniClipart (Wu et al., 2024) adopt similar paradigms for sketches or clipart. Despite promising visual results, these methods are not tailored to structured SVGs with closed shapes, fills, and layered occlusions, and they often lack persistent object identity and explicit topological guarantees, leading to unstable deformations or shape collapse. Moreover, their reliance on iterative optimization at inference time incurs high computational cost and limits controllability.

## E.4. Declarative and CSS-Based SVG Animation

Another line of work generates declarative animation code (e.g., CSS/SMIL) for existing SVG elements. Keyframer (Tseng et al., 2024) employs LLMs to synthesize CSS animation rules, enabling rapid prototyping and easy integration with web standards. However, declarative frameworks typically operate on affine transforms and appearance attributes (e.g., opacity and color) rather than directly modifying geometry (W3C, 2019; SVG Working Group, 2025). This prevents path-level deformations and complex motions such as morphing or elastic bending, which require explicit edits to SVG path data (the d attribute) (MDN contributors, 2025).

## E.5. Multimodal Grounding and Rendering-Aware Feedback

Because VAnim must jointly interpret rendered appearance, SVG code, and natural-language instructions, it is related to work on multimodal grounding and hallucination mitigation. Recent studies show that MLLMs can suffer from modality bias, attention misallocation, hallucinated visual evidence, and reasoning drift (Zhang et al., 2026b;c; 2025; Xi et al., 2026a;b). These findings motivate our identification-first design: before emitting code, VAnim explicitly binds the instruction to persistent SVG elements and thus reduces the chance of unconstrained, visually unfaithful updates.

More broadly, grounded reasoning methods bind intermediate decisions to localized evidence, structured visual programs, semantic prototypes, or causal/verification signals (Lin et al., 2026b;a; Zhu et al., 2025; 2026a; Lin et al., 2026c; Zhu et al., 2026b; Feng et al., 2026b; Feng & Ge, 2025; Feng et al., 2026a). Similar concerns appear in process-level multimodal evaluation, video understanding, online temporal perception, audio-visual segmentation, and efficient VLM deployment (Shi et al., 2026b; Zhang et al., 2026a; Zhong et al., 2024; Ying et al., 2023; Zhong et al., 2026; Chen et al., 2026b; Li et al., 2026a). Although these works are not designed for SVG animation, they reinforce the importance of grounding visual generation in reliable evidence rather than relying on text priors alone. Structured-context selection in language reasoning offers a related principle (Zhang et al., 2024b): irrelevant structure should be filtered before generation, analogous to selecting instruction-relevant SVG nodes before predicting sparse updates.

Rendering-aware reinforcement learning provides another complementary signal. Recent RL-based reasoning and code-generation systems use planning, reflection, execution feedback, or task-specific rewards to shape intermediate behavior (Wan et al., 2025b; Dou et al., 2025b; Li et al., 2026b; Dou et al., 2025a; Li et al., 2026c). Work on RL stability and distribution shift further highlights the need for well-conditioned objectives and localized updates (Qiao et al., 2026c;a;b; Wang et al., 2026a). VAnim follows this direction by optimizing sparse SVG updates with both rendered semantic alignment and format-validity rewards.

Finally, several lines of work address the efficiency and controllability of long structured generation. Long-context memory/computation methods and visual state-space models improve model-side efficiency or persistent state modeling (Wu et al., 2026; Wang, 2026; Wang & Xia, 2026; Shi et al., 2026a; Ke et al., 2025; 2026), while symbolic or structure-preserving

generation uses domain structure to improve controllability and preserve topology (Wang et al., 2026b; Chen et al., 2026a; Yuan et al., 2025a; Lu et al., 2026). These works are conceptually compatible with VAnim, but our contribution is specific to vector animation: we reduce the output-side sequence burden by preserving the SVG DOM and generating only sparse, editable state changes.

Overall, existing approaches do not jointly support persistent object identities, sparse state updates, and geometry-level deformation for vector animation, motivating VAnim's combination of identification-first planning, sparse SVG state modeling, and rendering-aware reinforcement learning.

## F. Limitations

While VAnim advances the state of the art in vector animation, we acknowledge several limitations:

- **Persistent DOM Structure:** The current formulation assumes a persistent SVG DOM tree with stable identifiers, so it is less suitable for animations that require node insertion, deletion, or dynamic topology changes.

- **Domain Specificity:** The training and evaluation data are drawn from a single source family (Lottie-derived SVGs), so out-of-domain generalization to hand-authored SVGs or design-tool exports remains an open question.

- **Computational Overhead:** Rendering-aware RL adds non-trivial compute overhead because each update samples and renders multiple candidates per prompt, which may limit deployability in resource-constrained environments.

For societal impact and ethical considerations, please refer to the Impact Statement in the main paper.

