# OpenReview forum: "VAnim: Rendering-Aware Sparse State Modeling for Structure-Preserving Vector Animation"
_ICML.cc/2026/Conference — ICML 2026 regular_

### Official Review · Reviewer_Tbtp · 2026-03-05

**Soundness:** 3
**Presentation:** 3
**Significance:** 3
**Originality:** 3
**Overall Recommendation:** 4
**Confidence:** 3

**Summary:**

This paper proposes VAnim, a framework for structure-preserving text-to-SVG animation generation. The key idea is to reformulate animation generation as sparse updates over a persistent SVG DOM tree, enabling identity consistency and editable outputs across frames. The approach is supported by a new dataset (SVGAnim-134k) and a training pipeline combining supervised learning with rendering-aware reinforcement learning.

**Compliance With Llm Reviewing Policy:**

Affirmed.

**Final Justification:**

My concerns have been well addressed by the authors' rebuttal, and I increase my score from 3 to 4.

**Key Questions For Authors:**

See weaknesses.

**Limitations:**

The authors do not adequately discuss the limitations or potential negative societal impact of their work. VAnim’s structural consistency relies on the Sparse State Update design, which may not generalize to animations involving dynamic topology changes or complex interactions. In addition, the computational cost of long-context modeling and rendering-aware RL is not analyzed, which may limit practical applicability and scalability.

**Strengths And Weaknesses:**

Strengths
- The paper study to generate editable and structurally consistent vector animations rather than rasterized videos.
- The proposed Sparse State Updates formulation is intuitive and provides a clear mechanism for maintaining identity consistency across frames. The introduction of SVGAnim-134k is a valuable contribution that may benefit future research in vector graphics generation.
- The integration of rendering-aware rewards aligns training with perceptual animation quality.

Weaknesses
1. The paper claims structural guarantees such as identity preservation and topological consistency; however, these appear to stem from design choices rather than formally established guarantees.
2. Reporting additional metrics (such as perceptual metric SSIM) for SVG animation could help better understand the proposed dataset SVGAnim-134k and better assess the benefit of introducing RL in VAnim.
3. In Table 5, including comparisons with ground truth would provide clearer context for judging result quality.
4. The computational cost is not discussed, making it difficult to assess scalability.

---

> ### Author Rebuttal · Authors · 2026-03-30
>
> We thank the reviewer for the constructive feedback and for recognizing the value of the SSU formulation, the SVGAnim-134k dataset, and the rendering-aware training strategy. To keep the response concise, we summarize the most relevant new results below and provide the full anonymous table here:
> https://anonymous.4open.science/r/Rebuttal-9239/experiment.png
>
> | Method / Variant | InternVideo2 ↑ | mean_flow_mag | flow_tLPIPS ↓ | SSIM ↑ | SR ↑ |
> |---|---:|---:|---:|---:|---:|
> | **VAnim (Ours)** | **0.202** | **1.711** | **0.0117** | **0.9719** | **100%** |
> | FlipSketch (Bandyopadhyay \& Song, 2025) | 0.137 | 1.696 | 0.1575 | 0.6786 | 100% |
> | NO GRPO (SFT-only) | 0.187 | 1.512 | 0.1360 | 0.9756 | 95.2% |
> |   |   |   |   |   |   |
>
> **1. Structural guarantees and wording (W1)**
>
> Thank you for pointing out this distinction. Our preservation property mainly comes from the SSU representation design rather than from a separate formal theorem. Because updates are applied on a persistent SVG DOM tree, non-participating elements and the DOM hierarchy are preserved across frames by construction. We will revise the manuscript accordingly and avoid stronger wording that overstates the level of formal guarantee.
>
> **2. Additional perceptual metrics and the benefit of RL (W2)**
>
> To evaluate the results more comprehensively, we added SSIM for identity preservation, flow_tLPIPS for temporal smoothness, and mean_flow_mag for motion magnitude. As shown in the table above, VAnim achieves SSIM = 0.9719 and flow_tLPIPS = 0.0117, and substantially outperforms FlipSketch in identity preservation (0.9719 vs. 0.6786).
>
> These metrics also make the benefit of RL clearer. As shown in the table above, compared with the ****NO GRPO (SFT-only)**** variant, GRPO increases mean_flow_mag from 1.512 to 1.711, indicating that RL alleviates the conservative-motion bias and encourages larger motion. Consistently, Appendix Fig. S3 shows that the frequency of non-rigid path `d` manipulations increases from 21% to 24%, moving closer to the GT distribution (30%).
>
> **3. Comparisons with Ground Truth (W3)**
>
> We believe this comment refers to Figure 5 rather than Table 5. We agree that GT context would make the qualitative comparison clearer, and we will add GT sequences alongside generated results in the revision.
>
> **4. Computational cost and scalability (W4)**
>
> We will add a clearer computational-efficiency discussion in the revision. For RL, Stage II takes approximately 93 wall-clock hours on 8× H100 GPUs for 420 update steps.
>
> Regarding context scalability, SSU is the mechanism we use to make long-horizon SVG animation tractable. As shown in Fig. 3 of the paper, naive frame-by-frame serialization requires 86.0k tokens on average for 24-frame animations, whereas SSU reduces this to 9.2k tokens, corresponding to a 9.86× compression ratio. While sequence lengths vary across samples, this reduction substantially improves feasibility under our 25k-token training setting and is particularly important for longer or structurally complex animations.
>
> **5. Limitations and societal impact**
>
> We will expand the limitations discussion to make the current scope more explicit. In particular, the current SSU formulation assumes a persistent DOM tree and is less suited to cases involving dynamic topology changes, node insertion/deletion, or more complex interactive behaviors. We will also state more clearly that the training distribution currently comes from a single source.
>
> We will additionally add a brief societal-impact discussion. Like other animation-generation systems, VAnim could be misused to produce deceptive or misleading animated content, and we therefore view provenance / watermarking for vector animations as an important future direction.
>
> We hope these additions clarify the scope, evidence, and limitations of VAnim, and we thank you again for the helpful suggestions.

---

> > ### Author Rebuttal · Reviewer_Tbtp · 2026-04-01
> >
> > Thank you for the authors' detailed response. My concerns have been well addressed, and I will increase the score.

---

> > > ### Author Response · Authors · 2026-04-02
> > >
> > > Dear Reviewer Tbtp,
> > >
> > > We sincerely appreciate your decision to raise the score!
> > >
> > > Best regards,
> > >
> > > Authors of paper #7216

---

### Official Review · Reviewer_9WcJ · 2026-03-09

**Soundness:** 2
**Presentation:** 4
**Significance:** 3
**Originality:** 3
**Overall Recommendation:** 4
**Confidence:** 3

**Summary:**

The paper introduces VAnim, an LLM-based framework that rethinks SVG animation through Sparse State Updates (SSU). Unlike previous methods that struggle with topological stability or high latency, VAnim treats animation as incremental edits on a persistent SVG DOM tree. This not only ensures identity consistency but also significantly reduces the sequence length. I found the Rendering-Aware RL strategy particularly interesting, as it effectively aligns discrete code tokens with visual feedback. With the release of the SVGAnim-134k dataset and strong results on non-rigid motion, this work offers a timely and mathematically rigorous contribution to vector graphics and multimodal learning.

**Compliance With Llm Reviewing Policy:**

Affirmed.

**Final Justification:**

Since the author solved my concerns, I would like to raise my score.

**Key Questions For Authors:**

please refer to the weakness.

**Limitations:**

please refer to the weakness.

**Strengths And Weaknesses:**

## Strengths
### Presentation:
The paper is clearly written, well-structured,
### Novelty:
The Rendering-Aware RL strategy with GRPO is a novel application of reinforcement learning to SVG code generation, solving the non-differentiable rendering problem.
### Dataset：
The paper propose the first large-scale and high-quality dataset for vector animation, giveing this field a great contribution.

## Weakness
1. The paper uses multiple rewards, but lacks an ablation experiment to analyze the value of each reward.

2. Whether the input visual information is truly helpful, and specifically how it helps (such as maintaining geometric consistency), has not been verified.

3. The paper uses PE-Core cosine similarity as the primary semantic alignment metric, but fails to explain why PE-Core is a more appropriate metric for vector animation than other video-text alignment metrics. Furthermore, it would be beneficial to provide validation of this metric against human judgment for comparison, thus better verifying its effectiveness.

---

> ### Author Rebuttal · Authors · 2026-03-30
>
> We thank the reviewer for the constructive feedback and for recognizing the novelty of the SSU formulation, the rendering-aware RL strategy, and the value of SVGAnim-134k. To keep the response concise, we summarize the most relevant new results below and provide the full anonymous table here:
> https://anonymous.4open.science/r/Rebuttal-9239/experiment.png
>
> | Method / Variant | InternVideo2 ↑ | mean_flow_mag | flow_tLPIPS ↓ | SSIM ↑ | SR ↑ |
> |---|---:|---:|---:|---:|---:|
> | **VAnim (Ours)** | **0.202** | **1.711** | **0.0117** | **0.9719** | **100%** |
> | NO R_align Reward | 0.191 | 1.589 | 0.1330 | 0.9811 | 100% |
> | NO R_fmt Reward | 0.199 | 1.705 | 0.1020 | 0.9734 | 96.6% |
> | No Input Image (Text + Code Only) | 0.176 | 1.548 | 0.1650 | 0.9245 | 96.3% |
> | GPT-5.2 | 0.180 | 0.954 | 0.0148 | 0.9505 | 88.5% |
> |   |   |   |   |   |   |
>
> **1. Ablation on the reward mechanisms (W1)**
>
> As shown in the table above, compared with full VAnim, ****removing R_fmt**** reduces the success rate from 100% to 96.6%, showing that this term acts as a syntactic and structural guardrail during exploration. As also shown in the table above, ****removing R_align**** lowers InternVideo2 from 0.202 to 0.191 and mean_flow_mag from 1.711 to 1.589, indicating that the visual semantic reward is important for encouraging richer motion instead of minimal updates.
>
> Taken together, these two rewards play different roles: R_fmt improves executability and structural validity, while R_align improves semantic faithfulness and motion expressiveness.
>
> **2. The role of the input visual information (W2)**
>
> We also conducted an explicit ****ablation without the input image (“Text + Code Only”)****. As shown in the table above, compared with full VAnim, removing the input image reduces SSIM from 0.9719 to 0.9245 and worsens flow_tLPIPS from 0.0117 to 0.1650. This shows that the rendered image is not redundant: it provides geometric cues that are difficult to infer reliably from SVG code alone, especially for complex path d attributes and spatial relations.
>
> This is also consistent with our model design. The visual encoder and cross-modal attention allow the model to align rendered visual entities with SVG DOM nodes, which helps the planning stage ground the correct objects and helps the execution stage preserve identity and geometric consistency.
>
> **3. Why PE-Core, and how it relates to human judgment (W3)**
>
> We agree that the original submission did not justify the choice of PE-Core clearly enough. We use PE-Core as the primary semantic-alignment metric because this task is fundamentally about animation semantics rather than static frame quality: many prompts depend on temporal relations such as gradual change, sequential motion, or coordinated multi-object behavior. In this setting, a video-text encoder that models spatio-temporal dynamics is more appropriate than framewise image-text metrics, which cannot directly capture motion logic at the sequence level. This is particularly important for vector animation, where semantic success often depends on whether the motion pattern matches the prompt, rather than on photorealistic appearance alone. At the same time, we do not claim that PE-Core is universally superior to all video-text encoders; rather, we use it as a reasonable primary evaluator and complement it with InternVideo2 as an independent cross-check.
>
> To reduce dependence on any single evaluator, we added InternVideo2 as an independent metric. As shown in the table above, VAnim remains the strongest among the compared methods under InternVideo2, achieving 0.202 versus 0.180 for GPT-5.2. As reported in Appendix Table S1, VAnim also achieves the best human ratings in Visual Integrity (4.62/5), Motion Smoothness (4.48/5), and Instruction Following (4.55/5), substantially outperforming Gemini 3 Pro (3.42/3.85/3.68). We therefore view the automatic metrics and the user study as broadly consistent, while avoiding a stronger claim of formal correlation analysis in the current submission.
>
> We hope these additions clarify the role of each reward, the necessity of the visual input, and the rationale behind the evaluation metrics. We will incorporate these ablations and clarifications into the revision.

---

> > ### Author Rebuttal · Reviewer_9WcJ · 2026-04-02
> >
> > The author provided a thorough and detailed analysis that fully resolved my questions. I will consider raising my score.

---

> > > ### Author Response · Authors · 2026-04-02
> > >
> > > Dear Reviewer 9WcJ,
> > >
> > > Thank you again for your acknowledgement and for considering a score update.
> > >
> > > If there are any remaining concerns, we would be very happy to clarify them further. If not, and if you feel our rebuttal has adequately addressed your questions, we would be very grateful if you would consider updating your recommendation.
> > >
> > > Thank you again for your time and helpful feedback.
> > >
> > > Best regards,
> > >
> > > Authors of paper #7216

---

### Official Review · Reviewer_DyyQ · 2026-03-12

**Soundness:** 3
**Presentation:** 3
**Significance:** 3
**Originality:** 2
**Overall Recommendation:** 4
**Confidence:** 3

**Summary:**

This paper studies the problem of text-to-SVG animation generation, aiming to create structure-preserving vector animations from natural language instructions. The authors observe that existing methods either rely on optimization-based pipelines that break SVG topology or LLM-based approaches that only support rigid CSS/SMIL transformations. To address these limitations, the paper proposes VAnim, an LLM-based framework that models animation as Sparse State Updates (SSU) on a persistent SVG DOM tree instead of generating full frame sequences. The method introduces Identification-First Motion Planning to ground textual instructions to specific SVG entities and employs Rendering-Aware Reinforcement Learning with GRPO to align discrete code updates with visual motion quality. The authors also construct SVGAnim-134k, the first large-scale dataset for vector animation generation.

**Compliance With Llm Reviewing Policy:**

Affirmed.

**Final Justification:**

The authors have addressed the concerns raised in my review during the rebuttal phase, and I have increased my score accordingly.

**Key Questions For Authors:**

1. Besides semantic alignment, consider including metrics that reflect temporal smoothness and geometric consistency, such as motion consistency or frame-to-frame structural stability.

2. Test the model on additional SVG animation datasets or unseen SVG structures to assess how well the method generalizes beyond the proposed SVGAnim-134k dataset.

3. Analyze the impact of key hyperparameters such as the sampling group size and the KL coefficient (β), and investigate how different settings affect training stability and performance.

**Limitations:**

yes

**Strengths And Weaknesses:**

### Summary of Strengths

1. **Well-designed modeling paradigm.**
   The proposed Sparse State Update (SSU) formulation models animation as updates on a persistent SVG DOM tree rather than full-frame generation, which effectively reduces sequence length and helps preserve topology and object identity.

2. **Integration of reasoning and rendering-aware learning.**
   The combination of Identification-First Motion Planning and rendering-aware reinforcement learning allows the model to ground semantic instructions to specific SVG entities and optimize visual motion quality.

3. **New large-scale dataset and benchmark.**
   The introduction of SVGAnim-134k provides the first large-scale dataset for vector animation generation and establishes a useful benchmark for future research.

### Summary of Weaknesses

1. The primary automatic metric focuses on semantic alignment, which may not fully reflect animation quality, such as temporal smoothness or geometric fidelity.

2. The experiments are mainly conducted on the proposed dataset, and it remains unclear how well the method generalizes to other vector animation sources or unseen SVG structures.

3. The paper does not provide sufficient analysis of key hyperparameters in the RL stage, such as the sampling group size and the KL coefficient (β), which may affect training stability and final performance.

---

> ### Author Rebuttal · Authors · 2026-03-30
>
> We thank the reviewer for the constructive feedback and for recognizing the value of the SSU formulation, the reasoning + RL pipeline, and the SVGAnim-134k benchmark. To keep the response concise, we summarize the most relevant new results below and provide the full anonymous table here:
> https://anonymous.4open.science/r/Rebuttal-9239/experiment.png
>
> | Method / Variant | InternVideo2 ↑ | mean_flow_mag ↑ | flow_tLPIPS ↓ | SSIM ↑ | SR ↑ |
> |---|---:|---:|---:|---:|---:|
> | FlipSketch (Bandyopadhyay \& Song, 2025) | 0.137 | 1.696 | 0.1575 | 0.6786 | 100% |
> | GPT-5.2 | 0.180 | 0.954 | 0.0148 | 0.9505 | 88.5% |
> | Gemini 3 Pro | 0.182 | 0.804 | 0.0136 | 0.9634 | 86.2% |
> | **VAnim (Ours)** | **0.202** | **1.711** | **0.0117** | **0.9719** | **100%** |
> |   |   |   |   |   |   |
> | G=4 | 0.195 | 1.598 | 0.0128 | 0.9801 | 100% |
> | G=8 (VAnim Default) | 0.202 | 1.711 | 0.0117 | 0.9719 | 100% |
> | G=16 | 0.207 | 1.743 | 0.0112 | 0.9689 | 100% |
> |   |   |   |   |   |   |
>
> **1. Additional metrics for animation quality (W1)**
>
> To evaluate the results more comprehensively, we added three metrics beyond semantic alignment: SSIM for identity preservation / geometric consistency, flow_tLPIPS for temporal smoothness, and mean_flow_mag for motion magnitude. As shown in the table above, VAnim performs best overall among the compared methods, with InternVideo2 = 0.202, SSIM = 0.9719, and flow_tLPIPS = 0.0117. These additions make the evaluation less one-dimensional and better reflect animation quality beyond prompt matching.
>
> **2. Generalization to unseen SVG structures and other sources (W2)**
>
> We have not yet completed a controlled out-of-domain benchmark beyond SVGAnim-134k, which we plan to explore in our future work. However, VAnim is not tied to semantic Lottie-style IDs: the grounding mechanism depends mainly on visual appearance rather than ID names. In practice, a lightweight preprocessing step can assign persistent IDs to SVG group and path nodes before inference, which makes extension to other SVG sources technically plausible.
>
> **3. Analysis of key RL hyperparameters (W3)**
>
> We added an ablation on the GRPO group size G. As shown in the table above, increasing ****G from 4 to 16**** improves InternVideo2 from 0.195 to 0.207 and mean_flow_mag from 1.598 to 1.743, but slightly reduces SSIM from 0.9801 to 0.9689. We therefore use G = 8 as the best overall trade-off.
>
> For the KL coefficient beta, our current setting is beta = 0.01. Empirically, this value stabilizes early GRPO training and avoids policy collapse while still allowing sufficient exploration for non-rigid deformation. We will clarify this design choice more explicitly in the revision.
>
> We hope these additions make the current evidence, scope, and limitations clearer, and we thank you again for the helpful suggestions.

---

> > ### Author Rebuttal · Reviewer_DyyQ · 2026-04-03
> >
> > The authors have largely addressed my concerns through additional experiments.

---

> > > ### Author Response · Authors · 2026-04-04
> > >
> > > Dear Reviewer DyyQ,
> > >
> > > Thank you again for your thoughtful feedback and for taking the time to review our rebuttal. We are grateful that the additional experiments and clarifications helped address your concerns.
> > >
> > > If there are any remaining issues, we would be very happy to clarify them further.
> > >
> > > Thank you again for your time and constructive suggestions.
> > >
> > > Best regards,
> > >
> > > Authors of paper #7216

---

### Official Review · Reviewer_rvzz · 2026-03-12

**Soundness:** 3
**Presentation:** 3
**Significance:** 3
**Originality:** 3
**Overall Recommendation:** 4
**Confidence:** 4

**Summary:**

This paper introduces VAnim, an LLM-based framework for text-to-SVG animation generation. The core idea is to model animation as Sparse State Updates (SSU) on a persistent SVG DOM tree rather than generating full SVG code frame-by-frame. This reduces token consumption by ~9.8× while preserving topological structure by construction. The framework uses a two-stage pipeline: (1) Supervised Fine-Tuning with an "Identification-First Motion Planning" chain-of-thought that grounds text instructions to specific SVG element IDs before generating code edits, and (2) Rendering-Aware Reinforcement Learning via GRPO, where a video perception encoder (PE-Core) provides visual feedback to align discrete code updates with continuous motion semantics. The authors also contribute SVGAnim-134k, a dataset of 134k vector animations sourced from Lottie files with dual-stream annotations (user prompts + structure-bound CoT). Experiments compare against GPT-5.2, Gemini 3 Pro, and LiveSketch on a held-out 1k test set, showing improvements in semantic alignment (PE-Core cosine similarity) and success rate.

**Compliance With Llm Reviewing Policy:**

Affirmed.

**Ethical Review Concerns:**

N/A (not flagged).

**Final Justification:**

I maintain my recommendation of Weak Accept (4). The paper's core contribution, the Sparse State Update (SSU) formulation for SVG animation, is an elegant and well-motivated representation that compresses token usage by ~9.8× while preserving DOM structure by construction. The Identification-First Motion Planning CoT and the rendering-aware GRPO pipeline are sensible design choices that together form a coherent system. SVGAnim-134k fills a genuine gap as the first large-scale vector animation dataset.

The rebuttal addressed my main concerns satisfactorily. (1) On the critical reward-metric circularity issue (W1/W3), the authors added InternVideo2 as an independent metric along with SSIM, flow_tLPIPS, and mean_flow_mag, and VAnim retained its lead across all of them—this substantially reduces my concern that gains were PE-Core-specific. (2) The naive frame-by-frame baseline (W2) was the most important new experiment: the fine-tuned Qwen3-VL-8B without SSU dropped to 62.3% success rate with severely degraded flow_tLPIPS (0.207 vs. 0.012), convincingly isolating SSU's contribution from the fine-tuning data advantage. AniClipart and FlipSketch were also added. (3) The GRPO analysis (W4) now shows concrete evidence of motion-bias alleviation (mean_flow_mag 1.512→1.711, path d manipulation frequency 21%→24%), and the group-size ablation clarifies the quality-diversity tradeoff. (4) The authors appropriately agreed to qualify the topological isomorphism claim (W6) and committed to adding implementation details, limitations, and societal impact discussion (W5).

Two residual concerns temper my enthusiasm. First, generalization beyond Lottie-derived SVGs (W7) remains unvalidated—the authors acknowledge this and defer to future work, which is honest but means the practical scope is currently narrow. Second, while the new metrics are welcome, the evaluation is still conducted on a single in-distribution test set from the same source as training. These factors prevent a stronger recommendation. Nonetheless, the SSU idea is clean and reusable, the dataset is a valuable resource, and the experimental evidence after rebuttal is substantially more convincing. The strengths in originality (SSU formulation), significance (underexplored problem, reusable dataset), and soundness (validated with broader metrics and fairer baselines) outweigh the remaining limitations.

**Key Questions For Authors:**

1. **Reward-metric circularity (critical):** PE-Core is used as both the GRPO reward signal and the primary evaluation metric. Can you provide evaluation with an independent video-text encoder (e.g., InternVideo, LanguageBind, or even CLIP-based metrics) to demonstrate that improvements are not specific to the PE-Core embedding space? This would significantly change my assessment if the gains hold across metrics.

2. **Fair baseline comparison:** The proprietary LLMs (GPT-5.2, Gemini 3 Pro) are not fine-tuned on SVGAnim-134k. Could you provide results for a Qwen3-VL-8B model fine-tuned with naive frame-by-frame SVG generation (without SSU) on the same 123k training set? This would isolate the contribution of the SSU representation from the data/fine-tuning advantage. Without this, it is unclear whether SSU or simply domain-specific fine-tuning drives the improvements.

3. **Path validity after updates:** When the model modifies `d` attributes for non-rigid deformation, what percentage of generated path commands are geometrically valid (e.g., correct number of control points, non-degenerate curves)? Is there a mechanism to enforce structural consistency of the path geometry itself, beyond the binary renderability check in Rfmt?

4. **Generalization to non-Lottie SVGs:** Have you tested VAnim on SVGs from other sources (e.g., Figma exports, hand-authored SVGs, or SVGs from static generation models like StarVector)? If the input SVG lacks semantic IDs, does the framework degrade gracefully?

5. **RL training details:** How many GRPO iterations are run in Stage II? What is the wall-clock training cost? How does performance vary with group size G and the relative weighting of λ_align vs. λ_fmt?

**Limitations:**

The paper briefly mentions future work directions (interactive behaviors, multi-scene narratives) but does not discuss several important limitations: (a) the dependency on pre-existing element IDs in input SVGs; (b) the single-source data distribution (Flaticon Lottie files only); (c) potential misuse for generating deceptive animated content; (d) the computational cost of the GRPO stage (rendering G=8 candidates per sample). The authors should add a more explicit limitations section discussing generalization bounds and a brief societal impact statement.

**Strengths And Weaknesses:**

**Strengths**

**S1: The Sparse State Update formulation is a genuinely elegant contribution.** The observation that 85%+ of SVG syntax is static across frames, and the reformulation as differential updates Δ_t anchored to persistent DOM node IDs, is both simple and powerful. The 9.8× token compression (Figure 3) is convincing and directly addresses the context explosion problem that makes naive autoregressive SVG animation intractable. Crucially, this representation provides topological preservation as a structural guarantee rather than a learned property — unchanged elements literally cannot drift because they are never regenerated. This is a clean engineering insight with theoretical backing.

**S2: Well-designed dataset and pipeline.** SVGAnim-134k fills a genuine gap — there is no prior large-scale dataset for vector animation with linguistic annotations. The pipeline from Lottie → ID-anchored SVG DOM → sparse state extraction → dual-stream annotation is well-thought-out. The strict ID-consistency filter (discarding samples where CoT references non-existent DOM IDs) is a nice quality control measure. The dataset split design (123k SFT / 10k RL / 1k test) with the RL subset curated specifically for geometric complexity is sensible.

**S3: The Identification-First Motion Planning is well-motivated and empirically validated.** The CoT decomposition into entity identification (mapping visual descriptions to SVG IDs) followed by dynamic planning is a natural fit for this structured generation task. The ablation (Table 2) shows it contributes the largest single improvement (−0.026 semantic alignment when removed), and the qualitative ablation (Figure 6) convincingly shows grounding failures — manipulating the wrong entity entirely — when this component is absent.

**S4: The paper tackles a genuinely underexplored and practically relevant problem.** Text-to-SVG animation sits at the intersection of code generation, visual reasoning, and temporal coherence, and the paper correctly identifies that neither optimization-based methods (topological instability) nor general-purpose LLMs (affine-only transforms, syntax degradation) adequately address it. The focus on non-rigid deformations via direct path `d` attribute manipulation is a meaningful advance over CSS/SMIL-limited approaches.

**Weaknesses**
**W1: The evaluation is narrow and potentially circular.** The primary quantitative metric — PE-Core cosine similarity — is the same model used as the GRPO reward signal. This creates a direct optimization-evaluation circularity: the model is literally trained to maximize the metric it is evaluated on. The paper does not acknowledge this concern. While it is common to use CLIP-based metrics in generative evaluation, using the *exact same encoder* for both reward and evaluation is problematic. At minimum, the authors should evaluate with an independent metric (e.g., a different video-text model, or decomposed metrics for motion magnitude, temporal smoothness, and semantic correctness separately). The user study (Table S1, deferred to supplementary) partially addresses this, but its details are not available for review in the main paper.

**W2: The baseline comparison is weak and incomplete.** The paper compares against only three baselines: GPT-5.2 (proprietary, general-purpose), Gemini 3 Pro (proprietary, general-purpose), and LiveSketch (optimization-based, designed for sketches not professional SVGs). Several important comparisons are missing: (a) AniClipart (Wu et al., 2024), which is directly relevant as it animates clipart using video priors; (b) FlipSketch (Bandyopadhyay & Song, 2025), cited in the references but not compared against; (c) concurrent SVG generation methods like Chat2SVG (Wu et al., 2025) or ReasonSVG (Xing et al., 2025a), which also use RL for SVG tasks. Furthermore, comparing a fine-tuned domain-specific 8B model against general-purpose LLMs used zero-shot (or few-shot) is not a fair comparison — GPT-5.2 and Gemini are not fine-tuned on SVGAnim-134k. A fairer comparison would include fine-tuning an open-source LLM (e.g., Qwen3-VL-8B without SSU, using naive frame-by-frame generation) to isolate the contribution of the SSU representation from the fine-tuning data advantage.

**W3: The quantitative evaluation lacks depth and statistical rigor.** Only two metrics are reported: PE-Core similarity (a single scalar) and success rate (binary). For an animation generation task, this is insufficient. Important missing metrics include: (a) temporal consistency / smoothness measures; (b) identity preservation scores (e.g., SSIM or LPIPS between the first frame and static elements in subsequent frames); (c) motion magnitude or diversity (to detect the "conservative motion bias" the paper itself identifies in the SFT ablation); (d) per-category breakdown across UI icons, loading indicators, and narrative illustrations. No confidence intervals or statistical significance tests are reported, despite the test set being only 1k samples.

**W4: The GRPO improvement is modest and underanalyzed.** The semantic alignment gain from SFT to GRPO is 0.268 → 0.281, a +0.013 improvement. While the success rate improvement (95.2% → 100%) is meaningful, the semantic alignment gain is small and its significance is unclear without variance estimates. The paper claims GRPO enables "complex non-rigid deformations beyond the reach of code-only supervision," but this claim is supported only by two qualitative examples in Figure 6 (conservative motion). A more systematic analysis — e.g., stratifying results by animation complexity, measuring the frequency and magnitude of path `d` attribute changes before and after RL — would substantiate this claim.

**W5: Key implementation details are missing or deferred.** Several important details that affect reproducibility are absent from the main paper: (a) The user study methodology and results (Table S1) are entirely in the supplementary; (b) The serialized SSU format (Figure S2) is also deferred; (c) The specific prompt templates used for Doubao-Seed-1.6 annotation are not provided; (d) The GRPO training details are sparse — how many RL iterations? What is the wall-clock training time for Stage II? How sensitive is performance to the group size G=8 and temperature T=0.9?

**W6: The "topological isomorphism" guarantee is overstated.** The paper repeatedly claims the SSU formulation "mathematically guarantees topological isomorphism." However, this guarantee only holds for elements that are NOT updated. For elements whose attributes ARE modified (particularly the path `d` attribute), there is no structural guarantee — an LLM could generate invalid path data, change the number of control points, or produce geometrically degenerate curves. The format validity reward Rfmt partially addresses this, but it checks renderability, not topological preservation of modified elements. The claim should be qualified: SSU guarantees preservation of *non-participating* elements and the *DOM tree structure*, but not the geometric validity of updated attributes.

**W7: Scalability and generalization concerns.** The method requires the input SVG to have pre-assigned, persistent element IDs — a property inherited from the Lottie-to-SVG conversion pipeline. Real-world SVGs from design tools (Figma, Illustrator) often have auto-generated, non-semantic IDs or lack IDs entirely. The paper does not discuss how VAnim would handle such inputs. Additionally, all training and evaluation is on SVGs derived from a single source (Flaticon Lottie files), raising concerns about generalization to other SVG styles.

---
**Soundness**

**3: good**

The core formulation (SSU + CoT + GRPO) is technically sound and the individual components are well-justified. The main soundness concern is the reward-metric circularity (W1) and the overstated topological guarantee (W6). The ablation study validates component contributions, though the evaluation methodology has gaps.

---

**Presentation**

**3: good**

The paper is generally well-written with a clear narrative arc. Figure 1 effectively showcases results, Figure 3 provides convincing token efficiency analysis, and Figure 4 gives a clear pipeline overview. The qualitative comparisons (Figure 5) are informative. However, deferring the user study and serialization format to the supplementary weakens the main paper. Some language is overly promotional ("mathematically guaranteeing," "rendering traditional temporal consistency constraints obsolete"). The related work coverage is thorough.

---

**Significance**

**3: good**

The problem is practically important and underexplored. SVGAnim-134k is a valuable community resource. The SSU formulation is a reusable idea that could generalize to other structured code generation tasks (e.g., HTML/CSS animation, 3D scene editing). However, the narrow evaluation and weak baselines limit confidence in the claimed significance of the specific quantitative results.

---

**Originality**

**3: good**

The SSU representation for SVG animation is novel and well-motivated. The combination of structure-bound CoT with GRPO visual rewards for bridging code generation and rendered output is a creative integration. However, individual components (CoT prompting, GRPO, PE-Core rewards) are established techniques, and the novelty lies primarily in their application to this specific domain and the SSU representation design.

---

**Key Questions For Authors**

---

> ### Author Rebuttal · Authors · 2026-03-30
>
> We thank the reviewer for the helpful feedback. We summarize the key results below and provide the full anonymous table here:
> https://anonymous.4open.science/r/Rebuttal-9239/experiment.png
>
> | Method / Variant | InternVideo2 ↑ | mean_flow_mag ↑ | flow_tLPIPS ↓ | SSIM ↑ | SR ↑ |
> |---|---:|---:|---:|---:|---:|
> | AniClipart (Wu et al., 2024) | 0.092 | 0.927 | 0.0376 | 0.9278 | 100% |
> | FlipSketch (Bandyopadhyay \& Song, 2025) | 0.137 | 1.696 | 0.1575 | 0.6786 | 100% |
> | GPT-5.2 | 0.180 | 0.954 | 0.0148 | 0.9505 | 88.5% |
> | Gemini 3 Pro | 0.182 | 0.804 | 0.0136 | 0.9634 | 86.2% |
> | LiveSketch (Gal et al., 2024) | 0.107 | 0.801 | 0.0612 | 0.9000 | 100% |
> | **VAnim (Ours)** | **0.202** | **1.711** | **0.0117** | **0.9719** | **100%** |
> |   |   |   |   |   |   |
> | NO SSU (Naive Frame-by-Frame) | 0.161 | 1.412 | 0.2070 | 0.9448 | 62.3% |
> | NO GRPO (SFT-only) | 0.187 | 1.512 | 0.1360 | 0.9756 | 95.2% |
> |   |   |   |   |   |   |
> | G=4 | 0.195 | 1.598 | 0.0128 | 0.9801 | 100% |
> | G=8 (VAnim Default) | 0.202 | 1.711 | 0.0117 | 0.9719 | 100% |
> | G=16 | 0.207 | 1.743 | 0.0112 | 0.9689 | 100% |
> |   |   |   |   |   |   |
>
> **1. Reward-metric circularity and quantitative depth (W1, W3)**
>
> To reduce the circularity concern, we added InternVideo2 as an independent metric, together with SSIM, flow_tLPIPS, and mean_flow_mag. As shown in the table above, VAnim achieves the best performance across all metrics, with InternVideo2 = 0.202 and SSIM = 0.9719. This makes the evaluation less dependent on PE-Core and provides a more complete view of semantic alignment, identity preservation, temporal smoothness, and motion magnitude.
>
> **2. Isolating the contribution of SSU and strengthening baselines (W2)**
>
> Following your suggestion, we fine-tuned a Qwen3-VL-8B variant with naive frame-by-frame SVG generation on the same training data (the variant ****NO SSU****). As shown in the table above, compared with VAnim, NO SSU drops from 100% to 62.3% in SR, and its flow_tLPIPS degrades from 0.0117 to 0.2070. This helps isolate the contribution of SSU from the benefit of domain-specific fine-tuning data alone.
>
> This is consistent with the token-efficiency analysis in Sec. 2.2. For 24-frame animations, naive frame-by-frame serialization requires 86.0k tokens on average, whereas SSU reduces this to 9.2k tokens (9.86× compression). This makes long-horizon SVG animation more tractable under our 25k-token setting, especially for longer or structurally complex examples.
>
> We also include AniClipart and FlipSketch. We do not include Chat2SVG (Wu et al., 2025) or ReasonSVG (Xing et al., 2025a) because they target static SVG generation rather than temporal animation.
>
> **3. GRPO improvement and additional analysis (W4)**
>
> By comparing the variant ****NO GRPO (SFT-only)**** with our full model, we can observe that GRPO increases mean_flow_mag from 1.512 to 1.711, indicating that RL alleviates the conservative-motion bias. Consistently, the frequency of non-rigid path `d` manipulations increases from 21% to 24%, moving closer to the GT distribution (30%). These results suggest that the visual reward encourages larger motion and more frequent non-rigid deformation, rather than minimal updates.
>
> The group-size ablation ****(G=4/8/16)**** shows that increasing G from 4 to 16 improves InternVideo2 from 0.195 to 0.207 and mean_flow_mag from 1.598 to 1.743, but slightly reduces SSIM from 0.9801 to 0.9689.
>
> **4. Overstated topological guarantee and path validity (W6)**
>
> SSU guarantees preservation of non-participating elements and the DOM tree structure; it does not provide a formal guarantee of geometric validity for edited path attributes. Our current safeguard for edited paths is executability and valid-ID checking via the format constraint/reward. We will tune down the claim accordingly.
>
> **5. Generalization beyond Lottie-style SVGs (W7)**
>
> We have not yet completed a controlled out-of-domain benchmark beyond SVGAnim-134k, which we plan to explore in our future work. However, VAnim is not tied to semantic Lottie-style IDs: the grounding mechanism depends mainly on visual appearance rather than ID names. In practice, a lightweight preprocessing step can assign persistent IDs to SVG group and path nodes before inference, which makes extension to other SVG sources technically plausible.
>
> **6. Additional implementation details and limitations (W5)**
>
> We will move key details from the supplementary material into the main paper, including the user study, the SSU serialization format, and the CoT prompt details. For RL, Stage II takes approximately 93 wall-clock hours on 8× H100 GPUs for 420 update steps. We will also make the limitations more explicit, including the dependence on persistent IDs / a stable DOM tree and the single-source training distribution, and add a brief societal-impact discussion on deceptive use.
>
> We hope these additions clarify the evidence, scope, and limitations of VAnim, and we thank you again for the detailed suggestions.

---

> > ### Author Rebuttal · Reviewer_rvzz · 2026-04-04
> >
> > My concerns have been addressed.

---

> > > ### Author Response · Authors · 2026-04-04
> > >
> > > Dear Reviewer rvzz,
> > >
> > > Thank you again for your thoughtful feedback and for taking the time to review our rebuttal.
> > >
> > > If there are any remaining concerns, we would be very happy to clarify them further. If not, and if you feel that our added experiments and clarifications have adequately addressed your questions, we would be very grateful if you would consider updating your recommendation.
> > >
> > > Thank you again for your time and helpful suggestions.
> > >
> > > Best regards,
> > >
> > > Authors of paper #7216

---

### Decision · Program_Chairs · 2026-04-30

**Decision:**

Accept (regular)

**Comment:**

VAnim reformulates text-to-SVG animation as Sparse State Updates (SSU) on a persistent SVG DOM tree, combining an Identification-First Motion Planning CoT with rendering-aware GRPO that uses a video-encoder reward. The paper also contributes SVGAnim-134k, the first large-scale vector animation dataset, and reports strong results against GPT-5.2, Gemini 3 Pro, LiveSketch, AniClipart, and FlipSketch.

**Strength**
- SSU is an elegant, well-motivated representation: ~9.8x token compression plus structural preservation of non-participating DOM elements by construction, tailored to the identity-drift problem in vector animation.
- Coherent system design — structure-bound CoT grounding + GRPO with a hybrid visual/format reward — with ablations showing each component (grounding, R_align, R_fmt, input image) contributes meaningfully.
- SVGAnim-134k fills a real gap as the first large-scale text-to-vector-animation dataset and is a reusable community resource.
- Rebuttal substantially strengthened the evaluation: added InternVideo2/SSIM/flow_tLPIPS/mean_flow_mag, added a crucial "no-SSU" fine-tuned baseline (SR drops 100% -> 62.3%), added AniClipart/FlipSketch, group-size ablation, and a user study — VAnim leads across all metrics.

**Weakness**
- Training and evaluation come from a single source (Flaticon/Lottie-derived SVGs); out-of-domain generalization (Figma/hand-authored SVGs, non-semantic IDs) is acknowledged but unvalidated.
- "Topological isomorphism" guarantee was overstated — holds for non-participating elements only, not for geometric validity of edited path-d attributes; authors agreed to qualify the claim.
- Individual components (CoT, GRPO, perception-encoder rewards) are established techniques; novelty rests primarily on the SSU formulation and its integration.

Overall, all four reviewers converged on Weak Accept post-rebuttal with concerns marked fully resolved; the SSU idea is clean and reusable, the dataset is valuable, and the expanded evaluation convincingly isolates the contribution.